# Deep Proxy Causal Learning and its Application to Confounded Bandit Policy Evaluation

**Liyuan Xu**
Gatsby Unit
liyuan.jo.19@ucl.ac.uk

**Heishiro Kanagawa**
Gatsby Unit
heishiro.kanagawa@gmail.com

**Arthur Gretton**
Gatsby Unit
arthur.gretton@gmail.com

## Abstract

Proxy causal learning (PCL) is a method for estimating the causal effect of treatments on outcomes in the presence of unobserved confounding, using *proxies* (structured side information) for the confounder. This is achieved via two-stage regression: in the first stage, we model relations among the treatment and proxies; in the second stage, we use this model to learn the effect of treatment on the outcome, given the context provided by the proxies. PCL guarantees recovery of the true causal effect, subject to identifiability conditions. We propose a novel method for PCL, the *deep feature proxy variable method (DFPV)*, to address the case where the proxies, treatments, and outcomes are high-dimensional and have nonlinear complex relationships, as represented by deep neural network features. We show that DFPV outperforms recent state-of-the-art PCL methods on challenging synthetic benchmarks, including settings involving high dimensional image data. Furthermore, we show that PCL can be applied to off-policy evaluation for the confounded bandit problem, in which DFPV also exhibits competitive performance.

## 1 Introduction

In causal learning, we aim to estimate the effect of our actions on the world. For example, we may be interested in measuring the impact of flight ticket prices on sales [2, 34], or the effect of grade retention on cognitive development [4]. We refer to our action as a *treatment*, which results in a particular *outcome*. It is often impossible to determine the effect of treatment on outcome from observational data alone, since the observed joint distribution of treatment and outcome can depend on a common *confounder* which influences both, and which might not be observed. In our example on sales of plane tickets given a price, the two might even be *positively correlated* in some circumstances, such as the simultaneous increase in sales and prices during the holiday season. This does not mean that raising the price *causes* sales to increase. In this context, people's desire to fly is a confounder, since it affects both the number of ticket sales and the prices people are willing to accept. Thus, we need to correct the bias caused by the confounder.

One common assumption to cope with confounding bias is to assume no unobserved confounders exist [8], or more generally, the *ignorable treatment assignment* assumption [28], which states that the treatment assignment is independent of the potential outcomes caused by the treatment, given the background data available. Although a number of methods are proposed based on this assumption [7, 9, 36], it can be too restrictive, since it is often difficult to determine how the confounder affects treatment assignments and outcomes.

35th Conference on Neural Information Processing Systems (NeurIPS 2021).

A less restrictive assumption is that we have access to *proxy variables*, which contain relevant side information on the confounder. In the flight tickets example, we can use the number of views of the ticket reservation page as a proxy variable, which reflects peoples' desire for flights. Note that if we can completely recover the confounder from proxy variables, the ignorable treatment assignment assumption can be satisfied. Motivated by this, Lee et al. [14] and Louizos et al. [17] aim to recover the distribution of confounders from proxy variables using modern machine learning techniques such as generative adversarial networks [5] or variational auto-encoders (VAE) [11]. Although these methods exhibit powerful empirical performance, there is little theory that guarantees the correct recovery of the causal effects.

Kuroki and Pearl [13] first considered the necessary conditions on proxy variables to provably recover the underlying causal effect via direct recovery of the hidden confounder. This work was in turn generalized by Miao et al. [20]. In their work, it is shown that two types of proxy variables are sufficient to recover the true causal effects *without* explicitly recovering the confounder. One is an *outcome-inducing proxy*, which *correlates with* confounders and *causes* the outcome, and the other is a *treatment-inducing proxy* which *is caused by* confounders and *correlates with* the treatment. In the flight ticket example, we can use the number of views of the ticket reservation page as the outcome-inducing proxy, and the cost of fuel as the treatment-inducing proxy. Given these proxy variables, Miao et al. [20] show that the true causal effect can be recovered by solving a Fredholm integral equation, which is referred to as the proxy causal learning (PCL) problem. The PCL problem is known to have an interesting relation to the causal inference with *multiple treatments*, which uses a subset of treatments as proxy variables in PCL [22, 33].

Although the PCL problem has a solid theoretical grounding, the question of how to estimate the causal effect remains a practical challenge, in particular when we consider nonlinear causal relationships or high-dimensional treatments. In Miao et al. [20], the treatment and outcome are assumed to be categorical variables. In a follow-up study, Miao et al. [21] show that we can learn a linear causal effect by a method of moments. Deaner [3] models the causal effect as a linear combination of nonlinear basis functions, which is learned by solving two stage regression. These two methods are extended by Mastouri and Zhu et al. [18], who learn the causal effect in a predefined reproducing kernel Hilbert space (RKHS). We provide an overview of the PCL problem and the two-stage regression in Section 2. Although these methods enjoy desirable theoretical properties, the flexibility of the model is limited, since all existing work uses pre-specified features.

In this paper, we propose a novel *Deep Feature Proxy Variable (DFPV)* method, which is the first work to apply neural networks to the PCL problem. The technique we employ builds on earlier work in *instrumental variable (IV) regression*, which is a related causal inference setting to PCL. A range of deep learning methods has recently been introduced for IV regression [1, 6, 35]. We propose to adopt the Deep Feature Instrumental Variable method [35], which learns deep adaptive features within a two-stage regression framework. Details of DFPV are given in Section 3. In Section 4, we empirically show that DFPV outperforms other PCL methods in several examples. We further apply PCL methods to the off-policy evaluation problem in a confounded bandit setting, which aims to estimate the average reward of a new policy given data with confounding bias. We discuss the setting in Section 3, and show the superiority of DFPV in experiments in Section 4.

## 2 Preliminaries

In this section, we introduce the proxy causal learning (PCL) problem and describe the existing two-stage regression methods to solve PCL.

**Notation.** Throughout the paper, a capital letter (e.g. $A$) denotes a random variable, and we denote the set where a random variable takes values by the corresponding calligraphic letter (e.g. $\mathcal{A}$). The symbol $\mathbb{P}(\cdot)$ denotes the probability distribution of a random variable (e.g. $\mathbb{P}(A)$). We use a lowercase letter to denote a realization of a random variable (e.g. $a$). We denote the expectation over a random variable as $\mathbb{E}[\cdot]$ and $\|f\|_{\mathbb{P}(\cdot)}$ as the $L^2$-norm of a function $f$ with respect to $\mathbb{P}(\cdot)$; i.e. $\|f\|_{\mathbb{P}(A)} = \sqrt{\mathbb{E}_A[f^2(A)]}$.

### 2.1 Problem Setting for Proxy Causal Learning

We begin with a description of the PCL setting. We observe a treatment $A \in \mathcal{A}$, where $\mathcal{A} \subset \mathbb{R}^{d_A}$, and the corresponding outcome $Y \in \mathcal{Y}$, where $\mathcal{Y} \subset \mathbb{R}$. We assume that there exists an unobserved confounder $U \in \mathcal{U}$ that affects both $A$ and $Y$. The goal of PCL is to estimate the structural function $f_{\text{struct}}(X)$ defined as

$$f_{\text{struct}}(a) = \mathbb{E}_U\left[\mathbb{E}_Y\left[Y|A=a,U\right]\right],$$

which we assume to be continuous. This function is also known as the *Average Treatment Effect (ATE)*. The challenge of estimating $f_{\text{struct}}$

Figure 1: Causal Graph.

is that the confounder $U$ is not observable — we cannot estimate the structural function from observations $A$ and $Y$ alone. To deal with this, we assume access to a treatment-inducing proxy variable $Z$, and an outcome-inducing proxy variable $W$, which satisfy the following *structural assumption* and *completeness assumption*.

**Assumption 1** (Structural Assumption [3, 18]). *We assume $Y \perp\!\!\!\perp Z|A,U$, and $W \perp\!\!\!\perp (A,Z)|U$.*

**Assumption 2** (Completeness Assumption on Confounder [3, 18]). *Let $l : \mathcal{U} \to \mathbb{R}$ be any square integrable function $\|l\|_{\mathbb{P}(U)} < \infty$. The following conditions hold:*

$$\mathbb{E}\left[l(U) \mid A=a, W=w\right] = 0 \quad \forall (a,w) \in \mathcal{A} \times \mathcal{W} \quad \Leftrightarrow \quad l(u) = 0 \quad \mathbb{P}(U)\text{-a.e.}$$
$$\mathbb{E}\left[l(U) \mid A=a, Z=z\right] = 0 \quad \forall (a,z) \in \mathcal{A} \times \mathcal{Z} \quad \Leftrightarrow \quad l(u) = 0 \quad \mathbb{P}(U)\text{-a.e.}$$

Figure 1 shows the causal graph describing these relationships. In our setting, we assume that there is no observable confounder, though this may be easily included [18, 31] as presented in Appendix D. Here, the bidirectional arrows mean that we allow both directions or even a common ancestor variable. Given these assumptions, it is shown that the structural function can be expressed using a solution of an integral equation.

**Proposition 1** (Miao et al. [20]). *Let Assumptions 1, 2 and Assumptions 4, 5, 6 in Appendix B hold. Then there exists at least one solution to the functional equation*

$$\mathbb{E}\left[Y|A=a, Z=z\right] = \int h(a,W)\rho_W(w|A=a, Z=z)\mathrm{d}w, \tag{1}$$

*which holds for any $(a,z) \in \mathcal{A} \times \mathcal{Z}$. Here, we denote $\rho_W(w|A=a, Z=z)$ as the density function of the conditional probability $\mathbb{P}(W|A=a, Z=z)$. Let $h^*$ be a solution of (1). The structural function $f_{\text{struct}}$ is given as*

$$f_{\text{struct}}(a) = \mathbb{E}_W\left[h^*(a,W)\right]. \tag{2}$$

For completeness, we present a proof in Appendix B (Lemma 2 and Corollary 1), which is due to Miao et al. [21] and Deaner [3]. Following Miao et al. [21], we call $h^*$ the *bridge function*. From Proposition 1, we can see that the estimation of the structural function reduces to the estimation of the bridge function, since once we obtain the bridge function, the structural function directly follows from (2).

## 2.2 Two-stage Regression Scheme for Proximal Causal Learning with Fixed Features

To obtain the bridge function, we need to solve the functional equation (1). However, directly solving (1) in a rich function space can be ill-posed (see discussion in Nashed and Wahba [25]). To address this, recent works [3, 18] minimize the following regularized loss $\mathcal{L}_{\text{PV}}$ to obtain an estimate of the bridge function $\hat{h}$:

$$\hat{h} = \arg\min_{h \in \mathcal{H}_h} \mathcal{L}_{\text{PV}}(h), \quad \mathcal{L}_{\text{PV}}(h) = \mathbb{E}_{Y,A,Z}\left[(Y - \mathbb{E}_{W|A,Z}\left[h(A,W)\right])^2\right] + \Omega(h), \tag{3}$$

where $\mathcal{H}_h$ is an arbitrary space of continuous functions and $\Omega(h)$ is a regularizer on $h$. Note that this loss can be interpreted as the deviation of the r.h.s and the l.h.s in (1) measured in $L^2$-norm with respect to the distribution $\mathbb{P}(Y,A,Z)$.

Deaner [3] and Mastouri and Zhu et al. [18] solve the minimization problem (3) by successively solving two-stage regression problems. They model the bridge function as

$$h(a,w) = \boldsymbol{u}^\top(\boldsymbol{\psi}_A(a) \otimes \boldsymbol{\psi}_W(w)) \tag{4}$$

where $\boldsymbol{u}$ is a learnable weight vector, $\boldsymbol{\psi}_A(a), \boldsymbol{\psi}_W(w)$ are vectors of fixed basis functions, and $\otimes$ is a Kronecker product, defined as $\boldsymbol{a} \otimes \boldsymbol{b} = \mathrm{vec}(\boldsymbol{ab}^\top)$ for any finite dimensional vectors $\boldsymbol{a}, \boldsymbol{b}$.[1] An estimate $\hat{\boldsymbol{u}}$ is obtained by solving the successive regression problems. In Stage 1, we estimate the conditional expectation $\mathbb{E}_{W|A=a,Z=z}[\boldsymbol{\psi}_W(W)]$ as a function of $a, z$. In Stage 2, we substitute the model (4) into the inner conditional expectation in $\mathcal{L}_{\mathrm{PV}}$,

$$\mathbb{E}_{W|A=a,Z=z}[h(a,W)] = \boldsymbol{u}^\top(\boldsymbol{\psi}_A(a) \otimes \mathbb{E}_{W|A=a,Z=z}[\boldsymbol{\psi}_W(W)]),$$

and then minimize $\mathcal{L}_{\mathrm{PV}}$ with respect to $\boldsymbol{u}$ using the estimate of $\mathbb{E}_{W|A=a,Z=z}[\boldsymbol{\psi}_W(W))]$ from Stage 1.

The above idea can be implemented as follows. We model the conditional expectation as

$$\mathbb{E}_{W|A=a,Z=z}[\boldsymbol{\psi}(W)] = \boldsymbol{V}(\boldsymbol{\phi}_A(a) \otimes \boldsymbol{\phi}_Z(z)),$$

where $\boldsymbol{\phi}_A(a), \boldsymbol{\phi}_Z(z)$ are another set of basis functions, and $\boldsymbol{V}$ is a *matrix* to be learned. Note that we can use different basis functions for $\boldsymbol{\phi}_A(a)$ and $\boldsymbol{\psi}_A(a)$. In Stage 1, the matrix $\boldsymbol{V}$ is learned by minimizing the following loss,

$$\mathcal{L}_1(\boldsymbol{V}) = \mathbb{E}_{W,A,Z}\left[\|\boldsymbol{\psi}_W(W) - \boldsymbol{V}(\boldsymbol{\phi}_A(A) \otimes \boldsymbol{\phi}_Z(Z))\|^2\right] + \lambda_1\|\boldsymbol{V}\|^2, \tag{5}$$

where $\lambda_1 > 0$ is a regularization parameter. This is a linear ridge regression problem with multiple targets, which can be solved analytically. In Stage 2, given the minimizer $\hat{\boldsymbol{V}} = \arg\min_{\boldsymbol{V}} \mathcal{L}_1(\boldsymbol{V})$, we can obtain $\hat{\boldsymbol{u}}$ by minimizing the loss

$$\mathcal{L}_2(\boldsymbol{u}) = \mathbb{E}_{Y,A,Z}\left[\|Y - \boldsymbol{u}^\top(\boldsymbol{\psi}_A(A) \otimes (\hat{\boldsymbol{V}}(\boldsymbol{\phi}_A(A) \otimes \boldsymbol{\phi}_Z(Z))))\|^2\right] + \lambda_2\|\boldsymbol{u}\|^2, \tag{6}$$

where $\lambda_2 > 0$ is another regularization parameter. Stage 2 corresponds to another linear ridge regression from input $\boldsymbol{\psi}_A(A) \otimes (\hat{\boldsymbol{V}}(\boldsymbol{\phi}_A(A) \otimes \boldsymbol{\phi}_Z(Z)))$ to target $Y$, and also enjoys a closed-form solution. Given the learned weights $\hat{\boldsymbol{u}} = \arg\min_{\boldsymbol{u}} \mathcal{L}_2(\boldsymbol{u})$, the estimated structural function is $\hat{f}_{\mathrm{struct}}(a) = \hat{\boldsymbol{u}}^\top(\boldsymbol{\psi}_A(a) \otimes \mathbb{E}_W[\boldsymbol{\psi}_W(W)])$.

When fixed feature dictionaries are used, this two-stage regression benefits from strong theoretical guarantees [3, 18]. The use of pre-specified feature maps, limits the scope and flexibility of the method, however, especially if the treatment and proxies are high dimensional (e.g. images or text), and the causal relations are nonlinear. To overcome this, we propose to use adaptive features, expressed by neural nets, as described in the next section.

## 3   Deep Feature Proxy Causal Learning

In this section, we develop the DFPV algorithm, which learns adaptive features modeled by neural nets using a technique similar to Xu et al. [35]. As in Mastouri and Zhu et al. [18], we assume that we do not necessarily have access to observations from the joint distribution of $(A, Y, Z, W)$. Instead, we are given $m$ observations of $(A, Z, W)$ for Stage 1 and $n$ observations of $(A, Z, Y)$ for Stage 2. We denote the stage 1 observations by $(a_i, z_i, w_i)$ and the stage 2 observations by $(\tilde{a}_i, \tilde{z}_i, \tilde{y}_i)$. If observations of $(A, Y, Z, W)$ are given for both stages, we can evaluate the out-of-sample loss of Stage 1 using Stage 2 data and vice versa, and these losses can be used for hyper-parameter tuning of $\lambda_1, \lambda_2$ (Appendix A). We first introduce two-stage regression with adaptive feature maps and then describe the detailed learning procedure of DFPV.

### 3.1   Two-stage regression with adaptive features

In DFPV, we consider the following models of the bridge function $h(a, w)$ and conditional feature mean $\mathbb{E}_{W|A,Z}[\boldsymbol{\psi}_{\theta_W}(W)]$:

$$h(a,w) = \boldsymbol{u}^\top(\boldsymbol{\psi}_{\theta_{A(2)}}(a) \otimes \boldsymbol{\psi}_{\theta_W}(w)), \quad \mathbb{E}_{W|A=a,Z=z}[\boldsymbol{\psi}_{\theta_W}(W)] = \boldsymbol{V}(\boldsymbol{\phi}_{\theta_{A(1)}}(a) \otimes \boldsymbol{\phi}_{\theta_Z}(z)),$$

where $\boldsymbol{u}$ and $\boldsymbol{V}$ are parameters, and $\boldsymbol{\phi}_{\theta_{A(1)}}, \boldsymbol{\phi}_{\theta_Z}, \boldsymbol{\psi}_{\theta_{A(2)}}, \boldsymbol{\psi}_{\theta_W}$ are neural nets parametrized by $\theta_{A(1)}, \theta_Z, \theta_{A(2)}, \theta_W$, respectively. Again, we may use different neural nets in the treatment features $\boldsymbol{\phi}_{\theta_{A(1)}}$ and $\boldsymbol{\psi}_{\theta_{A(2)}}$.

---

[1]Throughout this paper, we assume the number of basis functions to be finite. Mastouri and Zhu et al. [18] consider an infinite number of basis function in a reproducing kernel Hilbert space, and use the definitions of the inner and Kronecker products for that space.

As in the existing work [3, 18], we learn $\mathbb{E}_{W|a,z}[\psi_{\theta_W}(w)]$ in Stage 1 and $h(a,w)$ in Stage 2, but in addition to the weights $\boldsymbol{u}$ and $\boldsymbol{V}$, we also learn the parameters of the feature maps. Specifically, in Stage 1, we learn $\boldsymbol{V}$ and parameters $\theta_{A(1)}, \theta_Z$ by minimizing the following empirical loss:

$$\hat{\mathcal{L}}_1(\boldsymbol{V}, \theta_{A(1)}, \theta_Z) = \frac{1}{m}\sum_{i=1}^{m}\left\|\psi_{\theta_W}(w_i) - \boldsymbol{V}\left(\phi_{\theta_{A(1)}}(a_i)\otimes\phi_{\theta_Z}(z_i)\right)\right\|^2 + \lambda_1\|\boldsymbol{V}\|^2.$$

Note that $\hat{\mathcal{L}}_1$ is an empirical estimate of $\mathcal{L}_1$ in (5) with adaptive feature maps. Although $\hat{\mathcal{L}}_1$ depends on $\theta_W$, at this stage, we do not optimize $\theta_W$ with $\hat{\mathcal{L}}_1$ as $\psi_{\theta_W}(w)$ is the "target variable" in Stage 1. Given the minimizers $(\hat{\boldsymbol{V}}, \hat{\theta}_{A(1)}, \hat{\theta}_Z) = \arg\min \hat{\mathcal{L}}_1$, we learn weights $\boldsymbol{u}$ and parameters $\theta_W, \theta_{A(2)}$ by minimizing the empirical stage 2 loss,

$$\hat{\mathcal{L}}_2(\boldsymbol{u}, \theta_W, \theta_{A(2)}) = \frac{1}{n}\sum_{i=1}^{n}\left(\tilde{y}_i - \boldsymbol{u}^\top\left(\psi_{\theta_{A(2)}}(\tilde{a}_i)\otimes\hat{\boldsymbol{V}}\left(\phi_{\hat{\theta}_{A(1)}}(\tilde{a}_i)\otimes\phi_{\hat{\theta}_Z}(\tilde{z}_i)\right)\right)\right)^2 + \lambda_2\|\boldsymbol{u}\|^2.$$

Again, the loss $\hat{\mathcal{L}}_2$ is an empirical estimate of $\mathcal{L}_2$ in (6) with adaptive feature maps. Although the expression of $\hat{\mathcal{L}}_2$ does not explicitly contain $\theta_W$, it implicitly depends on $\theta_W$ as $(\hat{\boldsymbol{V}}, \hat{\theta}_{A(1)}, \hat{\theta}_Z)$ is the solution of a minimization problem involving $\theta_W$. This implicit dependency makes it challenging to update $\theta_W$, as we cannot directly obtain its gradient. One possible solution is to use the implicit gradient method [16], but this approach might be too computationally expensive. Instead, we use the method proposed in Xu et al. [35], in which we ignore the dependency of $\theta_W$ on parameters $\hat{\theta}_{A(1)}, \hat{\theta}_Z$, and compute the gradient via the closed-form solution of $\hat{V}$.

## 3.2 Deep Feature Proxy Variable Method

We now describe the learning procedure for DFPV. First, we fix parameters in the adaptive feature maps $(\theta_{A(1)}, \theta_Z, \theta_{A(2)}, \theta_W)$. Given these parameters, optimal weights $\hat{\boldsymbol{V}}, \hat{\boldsymbol{u}}$ can be learned by minimizing the empirical stage 1 loss $\hat{\mathcal{L}}_1$ and empirical stage 2 loss $\hat{\mathcal{L}}_2$, respectively. These minimizations can be solved analytically, where the solutions are

$$\hat{\boldsymbol{V}}(\boldsymbol{\theta}) = \Psi_1^\top\Phi_1(\Phi_1^\top\Phi_1 + m\lambda_1 I)^{-1}, \qquad \hat{\boldsymbol{u}}(\boldsymbol{\theta}) = \left(\Phi_2^\top\Phi_2 + n\lambda_2 I\right)^{-1}\Phi_2^\top\boldsymbol{y}_2, \qquad (7)$$

where we denote $\boldsymbol{\theta} = (\theta_{A(1)}, \theta_Z, \theta_{A(2)}, \theta_W)$ and define matrices as follows:

$$\Psi_1 = [\psi_{\theta_W}(w_1), \ldots, \psi_{\theta_W}(w_m)]^\top, \quad \Phi_1 = [\boldsymbol{v}_1(a_1, z_1), \ldots, \boldsymbol{v}_1(a_m, z_m)]^\top,$$
$$\boldsymbol{y}_2 = [\tilde{y}_1, \ldots, \tilde{y}_n]^\top, \qquad\qquad \Phi_2 = [\boldsymbol{v}_2(\tilde{a}_1, \tilde{z}_1), \ldots, \boldsymbol{v}_2(\tilde{a}_n, \tilde{z}_n)]^\top,$$
$$\boldsymbol{v}_1(a, z) = \phi_{\theta_{A(1)}}(a)\otimes\phi_{\theta_Z}(z), \qquad \boldsymbol{v}_2(a, z) = \psi_{\theta_{A(2)}}(a)\otimes\left(\hat{\boldsymbol{V}}(\boldsymbol{\theta})\left(\phi_{\theta_{A(1)}}(a)\otimes\phi_{\theta_Z}(z)\right)\right).$$

Given these weights $\hat{\boldsymbol{u}}(\boldsymbol{\theta}), \hat{\boldsymbol{V}}(\boldsymbol{\theta})$, we can update feature parameters by a gradient descent method with respect to the residuals of the loss of each stage, while regrading $\hat{V}$ and $\hat{u}$ as functions of parameters. Specifically, we take the gradient of the losses

$$\hat{\mathcal{L}}_1^{\mathrm{DFPV}}(\boldsymbol{\theta}) = \frac{1}{m}\sum_{i=1}^{m}\left\|\psi_{\theta_W}(w_i) - \hat{\boldsymbol{V}}(\boldsymbol{\theta})\left(\phi_{\theta_{A(1)}}(a_i)\otimes\phi_{\theta_Z}(z_i)\right)\right\|^2 + \lambda_1\|\hat{\boldsymbol{V}}(\boldsymbol{\theta})\|^2,$$

$$\hat{\mathcal{L}}_2^{\mathrm{DFPV}}(\boldsymbol{\theta}) = \frac{1}{n}\sum_{i=1}^{n}\left(\tilde{y}_i - \hat{\boldsymbol{u}}(\boldsymbol{\theta})^\top\left(\psi_{\theta_{A(2)}}(\tilde{a}_i)\otimes\hat{\boldsymbol{V}}(\boldsymbol{\theta})\left(\phi_{\theta_{A(1)}}(\tilde{a}_i)\otimes\phi_{\theta_Z}(\tilde{z}_i)\right)\right)\right)^2 + \lambda_2\|\hat{\boldsymbol{u}}(\boldsymbol{\theta})\|^2,$$

where $\hat{\boldsymbol{V}}(\boldsymbol{\theta}), \hat{\boldsymbol{u}}(\boldsymbol{\theta})$ are given in (7). Given these losses, $(\theta_{A(1)}, \theta_Z)$ are minimized with respect to $\hat{\mathcal{L}}_1^{\mathrm{DFPV}}(\boldsymbol{\theta})$, and $(\theta_{A(2)}, \theta_W)$ are minimized with respect to $\hat{\mathcal{L}}_2^{\mathrm{DFPV}}(\boldsymbol{\theta})$. Finally, we take the empirical mean of $\psi_{\theta_W^{(t)}}$ based on additional output-proxy data $S_W = \{w_i^{\mathrm{extra}}\}_{i=1}^{n_W}$, which is used for estimating the structural function. Here, we assume access to $S_W$ for proving consistency results, but empirically, we can use stage 1 data to compute this mean.

The complete procedure is presented in Algorithm 1. Note that we may use any sophisticated gradient-based learning method to optimize, such as Adam [10]. As reported in Xu et al. [35], we observe that the learning procedure is stabilized by running several gradient descent steps on the stage 1 parameters $(\theta_{A(1)}, \theta_Z)$ before updating the stage 2 features $(\theta_{A(2)}, \theta_W)$. Furthermore, we

may use mini-batch updates, which sample subsets of the data at the beginning of each iteration and only use these subsamples to update the parameters.

---

**Algorithm 1** Deep Feature Proxy Causal Learning

---

**Input:** Stage 1 data $S_1 = \{a_i, z_i, w_i\}$, Stage 2 data $S_2 = \{\tilde{a}_i \tilde{z}_i, \tilde{y}_i\}$, Additional outcome-proxy data $S_W = \{w_i^{\text{extra}}\}$, Regularization parameters $(\lambda_1, \lambda_2)$. Initial values $\boldsymbol{\theta}^{(0)} = (\theta_{A(1)}^{(0)}, \theta_Z^{(0)}, \theta_{A(2)}^{(0)}, \theta_W^{(0)})$. Learning rate $\alpha$.

**Output:** Estimated structural function $\hat{f}_{\text{struct}}(a)$

1: $t \leftarrow 0$
2: **repeat**
3:     Compute $\hat{\boldsymbol{V}}(\boldsymbol{\theta}^{(t)}), \hat{\boldsymbol{u}}(\boldsymbol{\theta}^{(t)})$ in (7)
4:     Update parameters in features $\boldsymbol{\theta}^{(t+1)} \leftarrow (\theta_{A(1)}^{(t+1)}, \theta_Z^{(t+1)}, \theta_{A(2)}^{(t+1)}, \theta_W^{(t+1)})$ as follows

$$\theta_{A(1)}^{(t+1)} \leftarrow \theta_{A(1)}^{(t)} - \alpha \nabla_{\theta_{A(1)}} \hat{\mathcal{L}}_1^{\text{DFPV}}(\boldsymbol{\theta})|_{\boldsymbol{\theta}=\boldsymbol{\theta}^{(t)}}, \quad \theta_Z^{(t+1)} \leftarrow \theta_Z^{(t)} - \alpha \nabla_{\theta_Z} \hat{\mathcal{L}}_1^{\text{DFPV}}(\boldsymbol{\theta})|_{\boldsymbol{\theta}=\boldsymbol{\theta}^{(t)}}$$

$$\theta_{A(2)}^{(t+1)} \leftarrow \theta_{A(2)}^{(t)} - \alpha \nabla_{\theta_{A(2)}} \hat{\mathcal{L}}_2^{\text{DFPV}}(\boldsymbol{\theta})|_{\boldsymbol{\theta}=\boldsymbol{\theta}^{(t)}}, \quad \theta_W^{(t+1)} \leftarrow \theta_W^{(t)} - \alpha \nabla_{\theta_W} \hat{\mathcal{L}}_2^{\text{DFPV}}(\boldsymbol{\theta})|_{\boldsymbol{\theta}=\boldsymbol{\theta}^{(t)}}$$

5:     Increment counter $t \leftarrow t + 1$;
6: **until convergence**
7: Compute $\hat{\boldsymbol{u}}(\boldsymbol{\theta}^{(t)})$ from (7)
8: Compute mean feature for $W$ using stage 1 dataset: $\boldsymbol{\mu}_{\theta_W} \leftarrow \frac{1}{n} \sum \boldsymbol{\psi}_{\theta_W^{(t)}}(w_i^{\text{extra}})$

9: **return** $\hat{f}_{\text{struct}}(a) = (\hat{\boldsymbol{u}}^{(t)})^\top \left( \boldsymbol{\psi}_{\hat{\theta}_{A(2)}^{(t)}}(a) \otimes \boldsymbol{\mu}_{\theta_W} \right)$

---

### 3.3 Consistency of DFPV

In this section, we show that the DFPV method yields a consistent estimate of the bridge function. Our approach differs from the consistency result of Deaner [3] and Mastouri and Zhu et al. [18], which depend on the assumption that the true bridge function lies in a specific functional space (such as an RKHS [18]) or satisfies certain smoothness properties [3]. Since our features are adaptive, however, we instead build our result on a Rademacher complexity argument [24], which holds for a wide variety of hypothesis spaces. Furthermore, we do not assume that our hypothesis space contains the true bridge function, which makes this theory of independent interest.

To derive the excess risk bound, we need another completeness assumption.

**Assumption 3** (Completeness Assumption on Outcome-Inducing Proxy [3, 18])**.** *Let $l : \mathcal{W} \to \mathbb{R}$ be any square integrable function $\|l\|_{\mathbb{P}(W)} < \infty$, We assume the following condition:*

$$\mathbb{E}\left[l(W) \mid A = a, Z = z\right] = 0 \quad \forall(a, z) \in \mathcal{A} \times \mathcal{Z} \quad \Leftrightarrow \quad l(w) = 0 \quad \mathbb{P}(W)\text{-a.e.}$$

This assumption is not necessary for identification (Proposition 1), but we need it for connecting the two-stage loss and the deviation of the bridge function. Given this assumption, we have the following consistency result.

**Proposition 2.** *Let Assumption 3 and Assumptions 7 and 8 in Appendix C hold. Given stage 1 data $S_1 = \{(w_i, a_i, z_i)\}_{i=1}^m$, stage 2 data $S_2 = \{(\tilde{y}_i, \tilde{a}_i, \tilde{z}_i)\}_{i=1}^n$, and additional output-proxy data $S_W = \{w_i^{\text{extra}}\}_{i=1}^{n_W}$, then for the minimizer of $\hat{\mathcal{L}}_2$ denoted as $(\hat{\boldsymbol{u}}, \hat{\theta}_{A(2)}, \hat{\theta}_Z)$, we have*

$$\|f_{\text{struct}} - \hat{f}_{\text{struct}}\|_{\mathbb{P}(A)} \leq O_p\left( \sqrt{\kappa_1 + \hat{\mathfrak{R}}_{S_1}(\mathcal{H}_1) + \frac{1}{\sqrt{m}}} + \kappa_2 + \sqrt{\hat{\mathfrak{R}}_{S_2}(\mathcal{H}_2) + \frac{1}{\sqrt{n}} + \frac{1}{\sqrt{n_W}}} \right),$$

*where $\mathcal{H}_1, \mathcal{H}_2$ are the functional classes that are considered in each stage, comprising both the predictions and their associated losses, which are defined in (12) and (15), respectively; and $\hat{\mathfrak{R}}_{S_1}(\mathcal{H}_1), \hat{\mathfrak{R}}_{S_2}(\mathcal{H}_2)$ are their respective empirical Rademacher complexities; the quantities $\kappa_1, \kappa_2$ measure the misspecification in each regression, and are defined in (13) and (16), respectively.*

We formally restate Proposition 2 in Appendix C (Theorem 3), in which the definitions of hypothesis spaces and constants can be found. From this, we can see that if we correctly identify the hypoth-

esis space ($\kappa_1, \kappa_2 = 0$) and have vanishing Rademacher complexities ($\hat{\mathfrak{R}}_{S_1}(\mathcal{H}_1), \hat{\mathfrak{R}}_{S_2}(\mathcal{H}_2) \to 0$ in probability), then the estimated bridge function converges in probability to the true one. Note that the Rademacher complexity of certain neural networks is known; e.g., ReLU two-layer networks are known to have the complexity of order $O(\sqrt{1/n})$ [26]. However, it might be not straightforward to derive the Rademacher complexities of the specific classes of neural networks we use, because we are employing the outer product of two neural networks in (5) and (6), whose Rademacher complexity remains a challenging open problem. The analysis in Proposition 2 is similar to Xu et al. [35], with two important distinctions: first, the target $f_{\text{struct}}$ is estimated using the bridge function $h$ marginalized over one of its arguments $W$, but PCL learns the bridge function itself. Second, our result is more general and includes the misspecified case, which was not treated in the earlier work.

One limitation of our analysis is that we leave aside questions of optimization. As we previously discussed, $\hat{\mathcal{L}}_2$ does not explicitly include $\theta_W$, thus it is difficult to develop a consistency result that includes the optimization. We emphasize that Algorithm 1 does not guarantee to converge to the global minimizer, since we ignore the implicit dependency of $\theta_W$ on $\theta_{A(1)}, \theta_Z$ when calculating the gradient. As we will see in the Section 4, however, the proposed method outperforms competing methods with fixed feature dictionaries, which do not have this issue.

### 3.4 DFPV for Policy Evaluation

PCL methods can estimate not only the structural function, but also other causal quantities, such as Conditional Average Treatment Effect (CATE) [29] and Average Treatment Effect on Treated (ATT) [3, 29]. Beyond the causal context, [32] uses a PCL method to conduct the off-policy evaluation in a partially observable Markov decision process with discrete action-state spaces. Here, we present the application of PCL methods to bandit off-policy evaluation with continuous actions, and propose a method to leverage adaptive feature maps in this context.

In bandit off-policy evaluation, we regard the outcome $Y$ as the reward and aim to evaluate the value function of a given policy. Denote policy $\pi : \mathcal{C} \to \mathcal{A}$ and define the value function $v(\pi)$ as

$$v(\pi) = \mathbb{E}_{U,C}\left[\mathbb{E}\left[Y|A = \pi(C), U\right]\right],$$

where $C \in \mathcal{C}$ is a variable on which the policy depends. In the flight ticket sales prediction example, a company might be interested in predicting the effect of discount offers. This requires us to solve the policy evaluation with $C = A$ since a new price is determined based on the current price. Alternatively, if the company is planning to introduce a new price policy that depends on the fuel cost, we need to consider the policy evaluation with $C = Z$, since the fuel cost can be regarded as the treatment-inducing proxy variable. Here, we assume the policy $\pi$ to be deterministic, but this can be easily generalized to a stochastic policy. Although the value function $v$ contains the expectation over the latent confounder $U$, it can be estimated through a bridge function.

**Proposition 3.** *Assume that the true bridge function $h^*(a, w) : \mathcal{A} \times \mathcal{W} \to \mathbb{R}$ is jointly measurable. Suppose $C \perp\!\!\!\perp W | U$. Under Assumptions 1, 2 and Assumptions 4, 5, 6 in Appendix B, we have*

$$v(\pi) = \mathbb{E}_{C,W}\left[h^*(\pi(C), W)\right].$$

The proof can be found in Appendix B (Corollary 2). From Proposition 3, we can obtain the value function by taking the empirical average over the bridge function $h(\pi(C), W)$. In DFPV, we estimate the bridge function as $h(a, w) = (\hat{\boldsymbol{u}})^\top (\boldsymbol{\psi}_{\hat{\theta}_{A(2)}}(a) \otimes \boldsymbol{\psi}_{\hat{\theta}_W}(w))$ where $\hat{\boldsymbol{u}}, \hat{\theta}_{A(2)}, \hat{\theta}_W$ minimize $\hat{\mathcal{L}}_2$. Hence, given $n'$ observation of $(C, W)$ denoted as $\{\check{c}_i, \check{w}_i\}_{i=1}^{n'}$, we can estimate the value function $v(\pi)$ as

$$\hat{v}(\pi) = \frac{1}{n_C} \sum_{i=1}^{n'} \hat{\boldsymbol{u}}^\top \left(\boldsymbol{\psi}_{\hat{\theta}_{A(2)}}(\pi(\check{c}_i)) \otimes \boldsymbol{\psi}_{\hat{\theta}_W}(\check{w}_i)\right).$$

We derive the following consistency result for policy value evaluation in a bandit with confounding bias.

**Proposition 4.** *Let Assumption 3 and Assumptions 7, 8, 9 in Appendix C hold. Denote the minimizers of $\hat{\mathcal{L}}_2$ as $(\hat{u}, \hat{\theta}_{A(2)}, \hat{\theta}_Z)$. Given stage 1 data $S_1 = \{(w_i, a_i, z_i)\}_{i=1}^m$, stage 2 data*

$S_2 = \{(\tilde{y}_i, \tilde{a}_i, \tilde{z}_i)\}_{i=1}^n$, and data for policy evaluation $S_3 = \{(\check{c}_i, \check{w}_i)\}_{i=1}^{n'}$, we have

$$|v(\pi) - \hat{v}(\pi)| \leq O_p\left(\sqrt{\kappa_1 + \hat{\mathfrak{R}}_{S_1}(\mathcal{H}_1) + \frac{1}{\sqrt{m}}} + \kappa_2 + \sqrt{\hat{\mathfrak{R}}_{S_2}(\mathcal{H}_2) + \frac{1}{\sqrt{n}}} + \frac{1}{\sqrt{n'}}\right)$$

*where $\mathcal{H}_1, \mathcal{H}_2$ are the function classes that are considered in each stage, comprising both the predictions and their associated losses, which are defined in* (12) *and* (15)*, respectively; and their empirical Rademacher complexities are given by $\hat{\mathfrak{R}}_{S_1}(\mathcal{H}_1), \hat{\mathfrak{R}}_{S_2}(\mathcal{H}_2)$; the quantities $\kappa_1, \kappa_2$ measure the misspecification in each regression, which are defined in* (13) *and* (16)*, respectively.*

The formal statement of Proposition 4 can be found in Appendix C (Theorem 3). As in the structural function case, we can see that estimated value function converges in probability to the true one, provided that we have vanishing Rademacher complexities and there is no misspecification.

## 4 Experiments

In this section, we report the empirical performance of the DFPV method. First, we present the results of estimating structural functions; we design two experimental settings for low-dimensional treatments and high-dimensional treatments, respectively. Then, we show the result of applying PCL methods to the bandit off-policy evaluation problem with confounding. We include the results for problems considered in prior work in Appendix E. The experiments are implemented using PyTorch [27]. The code is included in the supplemental material. All experiments can be run in a few minutes on Intel(R) Xeon(R) CPU E5-2698 v4 2.20GHz.

**Experiments for Structural Function** We present two structural function estimation experiments. One is a demand design experiment based on a synthetic dataset introduced by Hartford et al. [6], which is a standard benchmark for the instrumental variable regression. Here, we modify the data generating process to provide a benchmark for PCL methods. We consider the problem of predicting sales $Y$ from ticket price $P$, where these are confounded by a potential demand $D \in [0, 10]$. To correct this confounding bias, we use the fuel price $(C_1, C_2)$ as the treatment-inducing proxy, which has an impact on price $P$, and the number of views of the ticket reservation page $V$ as the outcome-inducing proxy. Details of the data generation process can be found in Appendix F.1.

Our second structural function estimation experiment considers high-dimensional treatment variables. We test this using the dSprite dataset [19], which is an image dataset described by five latent parameters (shape, scale, rotation, posX and posY). The images are $64 \times 64 = 4096$-dimensional. Based on this, Xu et al. [35] introduced the causal experiment, where the treatment is each figure, and the confounder is posY. Inspired by this, we consider the PCL setting that learns the same structural functions with nonlinear confounding, which is not possible to handle in the instrumental variable setting. Specifically, the structural function $f_{\text{struct}}$ and outcome $Y$ are defined as

$$f_{\text{struct}}(a) = \frac{\|\boldsymbol{B}a\|_2^2 - 5000}{1000}, \quad Y = \frac{1}{12}(\text{posY} - 0.5)f_{\text{struct}}(A) + \varepsilon, \quad \varepsilon \sim \mathcal{N}(0, 0.5),$$

where each element of the matrix $\boldsymbol{B} \in \mathbb{R}^{10 \times 4096}$ was generated from $\text{Unif}(0.0, 1.0)$ and fixed throughout the experiment. We fixed the shape parameter to heart and used other parameters as the treatment-inducing proxy $Z$. We sampled another image that shared the same posY as treatment $A$, which is used as output-inducing proxy $W$. Details of data generation process can be found in Appendix F.2.

We compare the DFPV method to three competing methods, namely KPV [18], PMMR [18], and an autoencoder approach derived from the CEVAE method [17]. In KPV, the bridge function is estimated through the two-stage regression as described in Section 2, where feature functions are fixed via their kernel functions. PMMR also models the bridge function using kernel functions, but parameters are learned by moment matching. CEVAE is not a PCL method, however, it represents a state-of-the-art approach in correcting for hidden confounders using observed proxies. The causal graph for CEVAE is shown in Figure 2, and CEVAE uses a VAE [11] to recover the distribution of confounder $U$ from the "proxy" $Q$. We make two modifications to CEVAE to apply it in our setting. First, we include

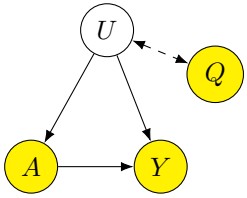

Figure 2: Causal Graph in CEVAE

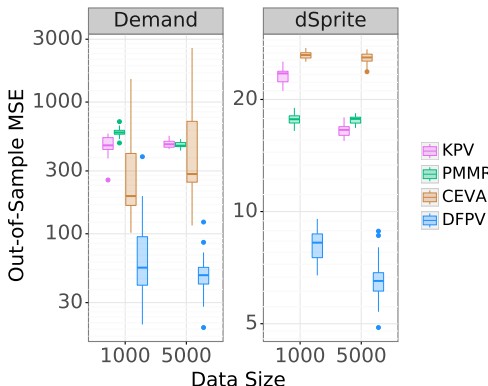

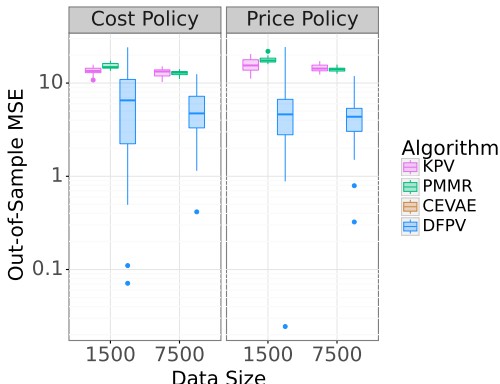

Figure 3: Result of structural function experiment in demand design setting (Left) and dSprite setting (Right).

Figure 4: Result of OPE experiment when the policy depends on the costs (Left) and on the current price (Right).

both the treatment-inducing proxy $Z$ and output-inducing proxy $W$ as $Q$ in CEVAE (we emphasize that this does not follow the causal graph in Figure 2, since there exist arrows from $Q$ to $A, Y$). Second, CEVAE is originally used in the setting where $Q$ is conditioned on a particular value, whereas we marginalize $Q$. See Appendix F.4 for the choice of the network structure and hyper-parameters. We tuned the regularizers $\lambda_1, \lambda_2$ as discussed in Appendix A, with the data evenly split for Stage 1 and Stage 2. We varied the dataset size and ran 20 simulations for each setting. Results are summarized in Figure 3.

In both experiments, DFPV consistently outperforms existing methods. This suggests that DFPV is capable of learning complex structural functions by taking the advantage of the flexibility of neural networks. KPV and PMMR perform similarly in all settings, but KPV tends to perform slightly better when the data size is large. This might be because KPV expresses its solution using a larger number of parameters than PMMR. Although CEVAE also learns a flexible model with a neural network, it's highly unstable in the demand design experiment and underperforms kernel methods in the dSprite experiment. This is because CEVAE does not take advantage of the relations between proxy variables and the structural function.

**Experiments for Offline Policy Evaluation** We now describe the offline policy evaluation experiment based on the demand design data. We set up synthetic experiments to evaluate two types of policy. In the first case, a policy depends on costs $C = (C_1, C_2)$, to address the question *How much would we gain if we decide the price based on the fuel costs*. In the second case, the policy depends on current action $C = P$, which addresses *How much would we gain if we cut the price by 30 percent*. The detailed policies can be found in Appendix F.3.

We only compare PCL methods here, as it is not straightforward to apply CEVAE to the off-policy evaluation problem. We evenly split the data for Stage 1, Stage 2, and policy evaluation (i.e we set $n = m = n'$).We ran 20 simulations for each setting. Results are summarized in Figure 4, in which DFPV performs better than existing PCL methods. This is not surprising since, as demonstrated in the structural function experiments, DFPV can estimate complex bridge functions, which results in a more accurate estimation of the value function.

## 5 Conclusion

We have proposed a novel approach for proxy causal learning, the Deep Feature Proxy Variable (DFPV) method, which performs two-stage least squares regression on flexible and expressive features. Motivated by the literature on the instrumental variable problem, we showed how to learn these feature maps adaptively with deep neural networks. We also showed that PCL learning can be used for off-policy evaluation in the bandit setting with confounding, and that DFIV performs competitively in this domain. This work thus brings together research from the worlds of deep offline RL and causality from observational data.

In future work, it would be interesting to explore different approaches to learning deep models in the PCL problem. One direction would be to adapt the method of moment matching, which has been studied extensively in the instrumental variable literature [1]. Moreover, Liao et al. [15] recently developed a novel adversarial method to solve a class of functional equations, which would be also a promising approach in the PCL problem. In the RL context, Tennenholtz et al. [32] shows an interesting connection between PCL learning and off-policy evaluation in a partially observable Markov decision process, and we believe that adapting DFPV to this setting will be of value.

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
