## A  Hyper-Parameter Tuning

If observations from the joint distribution of $(A, Y, Z, W)$ are available in both stages, we can tune the regularization parameters $\lambda_1, \lambda_2$ using the approach proposed in Singh et al. [30], Xu et al. [35]. Let the complete data of stage 1 and stage 2 be denoted as $(a_i, y_i, z_i, w_i)$ and $(\tilde{a}_i, \tilde{y}_i, \tilde{z}_i, \tilde{w}_i)$. Then, we can use the data not used in each stage to evaluate the out-of-sample performance of the other stage. Specifically, let Algorithm 1 converges at $t = T$, and the regularizing parameters are given by

$$\lambda_1^* = \arg\min \mathcal{L}_{1\text{-oos}}, \quad \mathcal{L}_{1\text{-oos}} = \frac{1}{n} \sum_{j=1}^{n} \left\| \boldsymbol{\psi}_{\theta_W^{(T)}}(\tilde{w}_j) - \hat{\boldsymbol{V}}^{(T)} \left( \boldsymbol{\phi}_{\theta_{A(1)}^{(T)}}(\tilde{a}_i) \otimes \boldsymbol{\phi}_{\theta_Z^{(T)}}(\tilde{z}_i) \right) \right\|^2,$$

$$\lambda_2^* = \arg\min \mathcal{L}_{2\text{-oos}},$$

$$\mathcal{L}_{2\text{-oos}} = \frac{1}{n} \sum_{i=1}^{n} \left( y_i - (\boldsymbol{u}^{(T)})^\top \left( \boldsymbol{\psi}_{\theta_{A(2)}^{(T)}}(a_i) \otimes \left( \hat{\boldsymbol{V}}^{(T)} \left( \boldsymbol{\phi}_{\theta_{A(1)}^{(T)}}(a_i) \otimes \boldsymbol{\phi}_{\theta_Z^{(T)}}(z_i) \right) \right) \right) \right)^2$$

where $\theta_Z^{(T)}, \theta_{A(1)}^{(T)}, \theta_W^{(T)}, \theta_{A(2)}^{(T)}, \hat{\boldsymbol{V}}^{(T)}, \boldsymbol{u}^{(T)}$ are the learned parameters by Algorithm 1.

## B  Identifiability

In this appendix, we prove propositions given in the main text. In the following, we assume that the spaces $\mathcal{U}, \mathcal{A}, \mathcal{Z}, \mathcal{W}$ are separable and completely metrizable topological spaces and equipped with Borel $\sigma$-algebras. In this section, we use the notation $P_{A|Z=z}$ to express the distribution of a random variable $A$ given another variable $Z = z$.

### B.1  Existence of bridge function

First, we discuss conditions to guarantee the existence of the bridge function $h$. Let us consider the following operators:

$$E_a : L^2(P_{W|A=a}) \to L^2(P_{Z|A=a}), \ E_a f := \mathbb{E}\left[ f(W)|A = a, Z = \cdot \right],$$

$$F_a : L^2(P_{Z|A=a}) \to L^2(P_{W|A=a}), \ F_a g := \mathbb{E}\left[ g(Z)|A = a, W = \cdot \right],$$

where the conditional expectations are identified as equivalent classes with natural inclusion into their corresponding $L^2$ spaces. Our goal is to show that $\mathbb{E}\left[Y|A = a, Z = \cdot\right]$ is in the range of $E_a$, i.e., we seek a solution of the inverse problem defined by

$$E_a h = \mathbb{E}\left[Y|A = a, Z = \cdot\right]. \tag{8}$$

This suffices to prove the existence of the function $h$, for if there exists a function $h_a^*$ for each $a \in \mathcal{A}$ such that

$$\mathbb{E}\left[h_a^*|A = a, Z = \cdot\right] = \mathbb{E}\left[Y|A = a, Z = \cdot\right],$$

we can define $h^*(a, w) := h_a^*(w)$. The existence of the bridge function corresponds to the results of Deaner [3, Lemma 1.1.b] and Miao et al. [20, Propisition 1]. For simplicity, in constrast to Deaner [3], we assume that there is no shared quantity between the proxy variables $W$ and $Z$. Following the proofs of the previous works [3, Appendix C, p. 57] and [20, Appendix 6], we aim to solve the integral equation in (8).

We will make use of the following theorem on the existence of a solution of a linear integral equation [12, Theorem 15.18].

**Proposition 5** ([12, Theorem 15.18])**.** *Let $\mathcal{X}$ and $\mathcal{Y}$ be Hilbert spaces. Let $E : \mathcal{X} \to \mathcal{Y}$ be a compact linear operator with singular system $\{(\mu_n, \varphi_n, g_n)\}_{n=1}^{\infty}$. The equation of the first kind*

$$E\varphi = f$$

*is solvable if and only if $f \in N(E^*)^{\perp}$ and*

$$\sum_{n=1}^{\infty} \frac{1}{\mu_n^2} |\langle f, g_n \rangle|^2 < \infty.$$

*Here, $N(E^*)$ denotes the null space of the operator $E^*$. Then a solution is given by*

$$\phi = \sum_{n=1}^{\infty} \frac{1}{\mu_n} \langle f, g_n \rangle \varphi_n.$$

To apply Proposition 5, we make the following additional assumptions.

**Assumption 4.** *For each $a \in \mathcal{A}$, the operator $E_a$ is compact with singular system $\{(\mu_{a,n}, \varphi_{a,n}, g_{a,n})\}_{n=1}^{\infty}$.*

**Assumption 5.** *For each $a \in \mathcal{A}$, the conditional expectation $f_{Y|a} := \mathbb{E}[Y|A = a, Z = \cdot]$ satisfies*

$$\sum_{n=1}^{\infty} \frac{1}{\mu_{a,n}^2} |\langle f_{Y|a}, g_{a,n} \rangle_{L^2(P_{Z|A=a})}|^2 < \infty,$$

*for a singular system $\{(\mu_{a,n}, \phi_{a,n}, g_{a,n})\}_{n=1}^{\infty}$ given in Assumption 4.*

**Remark 1.** *Assumption 4 is a minimal requisite to apply Proposition 5. The existing works [3, 20] assume stronger conditions; the operator $E_a$ is assumed to be Hilbert-Schmidt, which implies the compactness. Deaner [3, Assumption A.1] assumes that the joint distribution of $W$ and $Z$ conditioned on $a$ is absolutely continuous with respect to the product measure of $P_{W|A=a}$ and $P_{Z|A=a}$, and its density is square integrable. The density function serves as the integral kernel of the operator $E_a$ whose $L^2(P_{W|A=a} \otimes P_{Z|A=a})$-norm corresponds to the Hilbert-Schmidt norm of the operator, where we use $\otimes$ to denote the product measure. On the other hand, in the setting of Miao et al. [20], all probability distributions have densities with respect to the Lebesgue measure. The operator $E_a$ (denoted by $K_x$ in [20]) is defined by the relevant densities accordingly (see the paragraph after Lemma 2 of [20]). The compactness is, as in [3], established by the square-integrability of the integral kernel, which implies that the operator $E_a$ is Hilbert-Schmidt (see Condition A1 in [20]).*

It is easy to see that Assumptions 4 and 5 are required for using Proposition 5. The remaining condition to show is that $\mathbb{E}[Y|A = a, Z = \cdot]$ is in $N(E_a^*)^{\perp}$. We show that the structural assumption (Assumption 1) and completeness assumption (Assumption 2) imply the required condition. The result below closely follows [3] in that we share the same completeness condition.

**Lemma 1.** *Under Assumptions 1 and 2, the conditional expectation $\mathbb{E}[Y|A = a, Z = \cdot]$ is in the orthogonal complement of the null space $N(E_a^*)$.*

*Proof.* We first show that the adjoint of $E_a$ is given by $F_a$. For the operator $E_a$, any $f \in L_2(P_{W|A=a})$ and $g \in L_2(P_{Z|A=a})$, we have

$$\begin{aligned}
\langle E_a f, g \rangle_{L_2(P_{Z|A=a})} &= \mathbb{E}_{Z|A=a} \left[ \mathbb{E}[f(W)|A = a, Z] g(Z) \right] \\
&= \mathbb{E}_{Z|A=a} \left[ \mathbb{E}_{U|A=a,Z} \left[ \mathbb{E}[f(W)|A = a, Z, U] \right] g(Z) \right] \\
&\stackrel{(a)}{=} \mathbb{E}_{Z|A=a} \left[ \mathbb{E}_{U|A=a,Z} \left[ \mathbb{E}[f(W)|A = a, U] \right] g(Z) \right] \\
&= \mathbb{E}_{U,Z|A=a} \left[ \mathbb{E}[f(W)|A = a, U] g(Z) \right] \\
&= \mathbb{E}_{U|A=a} \left[ \mathbb{E}[f(W)|A = a, U] \mathbb{E}[g(Z)|A = a, U] \right]
\end{aligned}$$

where (a) follows from $W \perp\!\!\!\perp (A, Z)|U$, which is from Assumption 1. Similarly,

$$\begin{aligned}
\langle f, F_a g \rangle_{L^2(P_{W|A=a})} &= \mathbb{E}_{W|A=a} \left[ f(W) \mathbb{E}[g(Z)|A = a, W] \right] \\
&= \mathbb{E}_{W|A=a} \left[ f(W) \mathbb{E}_{U|A=a,W} \left[ \mathbb{E}[g(Z)|A = a, W, U] \right] \right] \\
&\stackrel{(b)}{=} \mathbb{E}_{W|A=a} \left[ f(W) \mathbb{E}_{U|A=a,W} \left[ \mathbb{E}[g(Z)|A = a, U] \right] \right] \\
&= \mathbb{E}_{W,U|A=a} \left[ f(W) \mathbb{E}[g(Z)|A = a, U] \right] \\
&= \mathbb{E}_{U|A=a} \left[ \mathbb{E}[f(W)|A = a, U] \mathbb{E}[g(Z)|A = a, U] \right] \\
&= \langle E_a f, g \rangle_{L_2(P_{Z|A=a})}.
\end{aligned}$$

Again, (b) is given by $W \perp\!\!\!\perp A, Z|U$ from Assumption 1. For any $f^* \in N(E_a^*) = N(F_a)$, by iterated expectations, we have

$$\begin{aligned}
0 &= \mathbb{E}[f^*(Z)|A = a, W = \cdot] \\
&= \mathbb{E}_U \left[ \mathbb{E}[f^*(Z)|A, U, W] | A = a, W = \cdot \right] \\
&= \mathbb{E}_U \left[ \mathbb{E}[f^*(Z)|A, U] | A = a, W = \cdot \right]. \tag{9}
\end{aligned}$$

From Assumption 2,

$$\mathbb{E}[l(U) \mid A = a, W = w] = 0 \quad \forall (a, w) \in \mathcal{A} \times \mathcal{W} \quad \Leftrightarrow \quad l(u) = 0 \quad P_U\text{-a.e.}$$

for all functions $l \in L^2(P_{U|A=a})$. Note that $\mathbb{E}\left[f^*(Z)|A,U\right] \in L^2(P_{U|A=a})$ since

$$\mathbb{E}_{U|A}\left[\mathbb{E}[f^*(Z)|A,U]^2\right] \leq \mathbb{E}_{U|A}\left[\mathbb{E}\left[f^*(Z)^2|A,U\right]\right] \quad (\because \text{Jensen's Inequality})$$
$$= \mathbb{E}\left[f^*(Z)^2|A\right] < \infty$$

Hence, (9) and Assumption 2 implies

$$\mathbb{E}\left[f^*(Z)|A=a, U=\cdot\right] = 0 \quad P_U\text{-a.s.}$$

Then, the inner product between $f^*$ and $\mathbb{E}\left[Y|A=a, Z=\cdot\right]$ is given as follows:

$$\langle f^*, \mathbb{E}\left[Y|A=a, Z=\cdot\right] \rangle_{L^2(P_{Z|A=a})} = \mathbb{E}_{Z|A=a}\left[f^*(Z)\mathbb{E}\left[Y|A=a, Z\right]\right]$$
$$= \mathbb{E}_{Z|A=a}\left[f^*(Z)\mathbb{E}_{U|A=a,Z}\left[\mathbb{E}\left[Y|A=a, Z, U\right]\right]\right]$$
$$\stackrel{(c)}{=} \mathbb{E}_{Z|A=a}\left[f^*(Z)\mathbb{E}_{U|A=a,Z}\left[\mathbb{E}\left[Y|A=a, U\right]\right]\right]$$
$$= \mathbb{E}_{U,Z|A=a}\left[f^*(Z)\mathbb{E}\left[Y|A=a, U\right]\right]$$
$$= \mathbb{E}_{U|A=a}\left[\mathbb{E}_{Z|U,A=a}\left[f^*(Z)\right]\mathbb{E}\left[Y|A=a, U\right]\right]$$
$$= 0.$$

Again (c) holds from $Y \perp\!\!\!\perp Z|A, U$ in Assumption 1. Hence, we have

$$\mathbb{E}\left[Y|A=a, Z=\cdot\right] \in N(E_a^*)^\perp.$$

$\square$

**Remark 2.** *The difference between the first condition in Assumption 2 and Condition 3 in [20] is in the approach to establishing that the conditional expectation $\mathbb{E}[Y|A=a, Z=\cdot]$ belongs to $N(F_a)^\perp$. More specifically, Condition 3 in [20] is equivalent to having $N(F_a) = \{0\}$ and so any nontrivial $L^2(P_{Z|A=a})$-function is in the orthogonal complement. Assumption 2 does not require $N(F_a) = \{0\}$ but only implies*

$$N(F_a) \subset \{f \in L^2(P_{Z|A=a}) : \mathbb{E}[f(Z)|A=a, U=\cdot] = 0\}.$$

*For our identifiability proof, we use the weaker condition, Assumption 2.*

Now, we are able to apply Proposition 5, leading to the following lemma.

**Lemma 2.** *Under Assumptions 1, 2, 4 and 5, for each $a \in \mathcal{A}$, there exists a function $h_a^* \in L_2(P_{W|A=a})$ such that*

$$\mathbb{E}\left[Y|A=a, Z=\cdot\right] = \mathbb{E}\left[h_a^*(W)|A=a, Z=\cdot\right].$$

*Proof.* By Lemma 1, the regression function $\mathbb{E}\left[Y|A=a, Z=\cdot\right]$ is in $N(E_a^*)^\perp$. Therefore, by Proposition 5, under the given assumptions, there exists a solution to (8). Letting the solution be $h_a^*$ completes the proof. $\square$

## B.2 Identifiability

Here, we show that the bridge function $h$ can be used to compute the various causal quantities. In addition to the assumptions required for the existence of bridge function listed in the previous section, we assume the conditional expectation of outcome given treatment is square integrable.

**Assumption 6.** *For each $a \in \mathcal{A}$, under the observational distribution, we have*

$$\mathbb{E}\left[Y^2|A=a\right] < \infty.$$

Given the assumption, we can prove the following theorem.

**Theorem 1** (Identifiability). *Assume that there exists a function $h : \mathcal{A} \times \mathcal{W} \to \mathbb{R}$ such that for each $a \in \mathcal{A}$, the function $h(a, \cdot)$ is in $L^2(P_{W|A=a})$ and satisfies*

$$\mathbb{E}\left[Y|A=a, Z=\cdot\right] = \mathbb{E}\left[h(a, W)|A=a, Z=\cdot\right] \quad P_{Z|A=a}\text{-a.s.}$$

*Under Assumptions 1, 2 and 6, we have*

$$\mathbb{E}_{W|U}\left[h(a, W)\right] = \mathbb{E}\left[Y|A=a, U\right]. \tag{10}$$

*Proof.* Note that we have

$$\mathbb{E}\left[h(a,W)|A=a,Z\right] = \mathbb{E}_{U|A=a,Z}\left[\mathbb{E}\left[h(a,W)|A=a,Z,U\right]\right]$$
$$= \mathbb{E}_{U|A=a,Z}\left[\mathbb{E}\left[h(a,W)|U\right]\right],$$

$$\mathbb{E}\left[Y|A=a,Z\right] = \mathbb{E}_{U|A=a,Z}\left[\mathbb{E}\left[Y|A=a,Z,U\right]\right]$$
$$= \mathbb{E}_{U|A=a,Z}\left[\mathbb{E}\left[Y|A=a,U\right]\right],$$

where the second line of each equation follows from Assumption 1. Moreover, by assumption, we have

$$\mathbb{E}_{U|A=a}\left[\mathbb{E}\left[h(a,W)|U\right]^2\right] \leq \mathbb{E}_{W|A=a}\left[h(a,W)^2\right] < \infty, \text{ and}$$

$$\mathbb{E}_{U|A=a}\left[\mathbb{E}\left[Y|A=a,U\right]^2\right] \leq \mathbb{E}\left[Y^2|A=a\right] < \infty.$$

Note from Assumption 2, we have

$$\mathbb{E}\left[l(U)\mid A=a,Z=z\right]=0 \quad \forall(a,z)\in\mathcal{A}\times\mathcal{Z} \quad \Leftrightarrow \quad l(u)=0 \quad P_U\text{-a.e.}$$

Therefore, by setting $l(u) = \mathbb{E}\left[Y|A=a,U=u\right] - \mathbb{E}\left[h(a,W)|U=u\right]$, for all $a\in\mathcal{A}$, we have

$$\mathbb{E}\left[h(a,W)|A=a,Z=\cdot\right] = \mathbb{E}\left[Y|A=a,Z=\cdot\right] \quad P_{Z|A=a}\text{-a.s.}$$
$$\Leftrightarrow \mathbb{E}_{U|A=a,Z=\cdot}\left[\mathbb{E}\left[h(a,W)|U\right] - \mathbb{E}\left[Y|A=a,U\right]\right] = 0 \quad P_{Z|A=a}\text{-a.s.}$$
$$\Leftrightarrow \mathbb{E}_{W|U=\cdot}\left[h(a,W)\right] = \mathbb{E}\left[Y|A=a,U=\cdot\right] \quad P_U\text{-a.s.}$$

$\square$

Using Theorem 1, we can show following two corollaries, which are used in the main body.

**Corollary 1.** *Let the assumptions in Theorem 1 hold. Given a bridge function $h^*$, we can estimate structural function $f_{\mathrm{struct}}$ as*

$$f_{\mathrm{struct}}(a) := \mathbb{E}_U\left[\mathbb{E}\left[Y|A=a,U\right]\right]$$
$$= \mathbb{E}_W\left[h^*(a,W)\right]$$

*Proof.* From Theorem 1, we have

$$\mathbb{E}_U\left[\mathbb{E}\left[Y|A=a,U\right]\right] = \mathbb{E}_U\left[\mathbb{E}_{W|U}\left[h^*(a,W)\right]\right] = \mathbb{E}_W\left[h^*(a,W)\right]$$

$\square$

**Remark 3.** *The above corollary corresponds to the identifiablity results in obtained in the previous works [3, 20]. We follow the proof of Theorem 1.1.a in [3, Appendix B] (See also, Theorem 1 and Appendix 3 of [20]).*

**Corollary 2.** *Assume we are given a bridge function $h^*(a,w)$ that is jointly measurable. Suppose $C \perp\!\!\!\perp W|U$. With the assumptions in Theorem 1, we can write the value function of policy $\pi(C)$ as*

$$v(\pi) := \mathbb{E}_{C,U}\left[\mathbb{E}\left[Y|A=\pi(C),U\right]\right]$$
$$= \mathbb{E}_C\left[\mathbb{E}_{W|C}\left[h(\pi(C),W)\right]\right]$$

*Proof.* From Theorem 1, we have

$$\mathbb{E}_{C,U}\left[\mathbb{E}\left[Y|A=\pi(C),U\right]\right] = \mathbb{E}_{C,U}\left[\mathbb{E}_{W|U}\left[h^*(\pi(C),W)\right]\right] = \mathbb{E}_C\left[\mathbb{E}_{W|C}\left[h(\pi(C),W)\right]\right],$$

where the last equality holds by the conditional independence $C \perp\!\!\!\perp W|U$. $\square$

Note that in the existence claim, the bridge function $h^*(a,w)$ is constructed by aggregating over $\{h_a^*\}_{a\in\mathcal{A}}$. This construction does not guarantee that the function is measurable with respect to $a$; the lack of measurability renders its expectation undefined. Thus, we have additionally assumed the joint measurability of the bridge function in Corollary 2. Validating this assumption is crucial theoretically, and we leave it for future work.

## C   Consistency of DFPV algorithm

In this appendix, we prove consistency of the DFPV approach. Following Xu et al. [35], we establish consistency of the end-to-end procedure incorporating Stages 1 and 2. We establish the result by first

showing a Stage 1 consistency result (Lemma 3), and then establishing the consistency of Stage 2 with the empirical Stage 1 solution used as input (Lemma 4). The desired result then follows in Theorem 2. Here, we assume the bridge function $h^*(a, w)$ is jointly measurable so that the expectation of $h^*$ is defined.

Consistency results will be expressed in terms of the complexity of the function classes used in Stages 1 and 2, as encoded in the Rademacher complexity of the function space of these functions (see Proposition 6 below). Consistency for particular function classes can then be shown by establishing that the respective Rademacher complexities vanish. We leave for future work the task of demonstrating this property for individual function classes of interest.

### C.1 Operator view of DFPV

The goal of DFPV is to learn a bridge function $h^*$, which satisfies

$$\mathbb{E}_{Y|A=a, Z=z}[Y] = \mathbb{E}_{W|A=a, Z=z}[h^*(a, W)]. \tag{11}$$

We use the model

$$h(a, w) = \boldsymbol{u}^\top (\boldsymbol{\psi}_{\theta_{A(2)}}(a) \otimes \boldsymbol{\psi}_{\theta_W}(w))$$

and denote the hypothesis spaces for $\boldsymbol{\psi}_{\theta_W}$ and $h$ as $\mathcal{H}_{\boldsymbol{\psi}_{\theta_W}} : \mathcal{W} \to \mathbb{R}^{d_W}$ and $\mathcal{H}_h : \mathcal{A} \times \mathcal{W} \to \mathbb{R}$, respectively. To learn the parameters, we minimize the following stage 2 loss:

$$\hat{\boldsymbol{u}}, \hat{\theta}_W, \hat{\theta}_{A(2)} = \underset{\boldsymbol{u}, \theta_W, \theta_{A(2)}}{\arg\min} \ \hat{\mathcal{L}}_2(\boldsymbol{u}, \theta_X, \theta_{A(2)}),$$

$$\hat{\mathcal{L}}_2 = \frac{1}{n} \sum_{i=1}^{n} (\tilde{y}_i - \boldsymbol{u}^\top (\boldsymbol{\psi}_{\theta_{A(2)}}(\tilde{a}_i) \otimes \hat{\mathbb{E}}_{W|A, Z}[\boldsymbol{\psi}_{\theta_W}(W)](\tilde{a}_i, \tilde{z}_i)))^2.$$

We denote the resulting estimated bridge function as

$$\hat{h}(a, w) = (\hat{\boldsymbol{u}})^\top (\boldsymbol{\psi}_{\hat{\theta}_{A(2)}}(a) \otimes \boldsymbol{\psi}_{\hat{\theta}_W}(w)).$$

For simplicity, we set all regularization terms to zero. Here, $\hat{\mathbb{E}}_{W|A, Z}[\boldsymbol{\psi}_{\theta_W}(W)]$ is the empirical conditional expectation operator, which maps an element of $\mathcal{H}_{\boldsymbol{\psi}_{\theta_W}}$ to some function $\boldsymbol{g}(a, z)$ which is defined as

$$\hat{\mathbb{E}}_{W|A, Z}[\boldsymbol{\psi}_{\theta_W}(W)] = \underset{\boldsymbol{g} \in \mathcal{G}}{\arg\min} \ \hat{\mathcal{L}}_1(\boldsymbol{g}; \boldsymbol{\psi}_{\theta_W}),$$

$$\hat{\mathcal{L}}_1 = \frac{1}{n} \sum_{i=1}^{m} \|\boldsymbol{\psi}_{\theta_W}(w_i) - \boldsymbol{g}(a_i, z_i)\|^2,$$

where $\|\cdot\|$ is the $\ell_2$-norm, and $\mathcal{G}$ is an arbitrary function space. In DFPV, we specify $\mathcal{G}$ as the set consisting of functions $\boldsymbol{g}$ of the form

$$\boldsymbol{g} = \boldsymbol{V}(\boldsymbol{\phi}_{\theta_{A(1)}}(a_i) \otimes \boldsymbol{\phi}_{\theta_Z}(z_i)).$$

Note that this formulation is equivalent to the one introduced in Section 3. With a slight abuse of notation, for $h(a, w) = \boldsymbol{u}^\top \boldsymbol{\psi}_{\theta_{A(2)}}(a) \otimes \boldsymbol{\psi}_{\theta_W}(w) \in \mathcal{H}_h$, we define $\hat{\mathbb{E}}_{W|A, Z}[h]$ to be

$$\hat{\mathbb{E}}_{W|A, Z}[h(A, W)](a, z) = \boldsymbol{u}^\top \left(\boldsymbol{\psi}_{\theta_{A(2)}}(a) \otimes \hat{\mathbb{E}}_{W|A, Z}[\boldsymbol{\psi}_{\theta_W}](a, z)\right)$$

since this is the empirical estimate of $\mathbb{E}_{W|A, Z}[h(A, W)]$.

### C.2 Generalization errors for regression

Here, we bound the generalization errors of both stages using Rademacher complexity bounds [24].

**Proposition 6.** *[Theorem 3.3 24, with slight modification] Let $\mathcal{S}$ be a measurable space and $\mathcal{H}$ be a family of functions mapping from $\mathcal{S}$ to $[0, M]$. Given fixed dataset $S = (s_1, s_2, \ldots, s_n) \in \mathcal{S}^n$, the empirical Rademacher complexity is given by*

$$\hat{\mathfrak{R}}_S(\mathcal{H}) = \mathbb{E}_{\boldsymbol{\sigma}} \left[ \frac{1}{n} \sup_{h \in \mathcal{H}} \sum_{i=1}^{n} \sigma_i h(s_i) \right],$$

*where $\boldsymbol{\sigma} = (\sigma_1, \ldots, \sigma_n)$, with $\sigma_i$ independent random variables taking values in $\{-1, +1\}$ with equal probability. Then, for any $\delta > 0$, with probability at least $1 - \delta$ over the draw of an i.i.d sample*

$S$ of size $n$, each of following holds for all $h \in \mathcal{H}$:

$$\mathbb{E}\left[h(s)\right] \leq \frac{1}{n}\sum_{i=1}^{n}h(s_i) + 2\hat{\mathfrak{R}}_S + 3M\sqrt{\frac{\log 2/\delta}{2n}},$$

$$\frac{1}{n}\sum_{i=1}^{n}h(s_i) \leq \mathbb{E}\left[h(s)\right] + 2\hat{\mathfrak{R}}_S + 3M\sqrt{\frac{\log 2/\delta}{2n}}.$$

We list the assumptions below.

**Assumption 7.** *The following hold:*

1. *Bounded outcome variable $|Y| \leq M$.*

2. *Bounded stage 1 hypothesis space: $\forall a \in \mathcal{A}, z \in \mathcal{Z}, \|\boldsymbol{g}(a,z)\| \leq 1$.*

3. *Bounded stage 2 feature map $\forall w \in \mathcal{W}, \|\boldsymbol{\psi}_{\theta_W}(w)\| \leq 1$.*

4. *Bounded stage 2 weight: $\forall \|\boldsymbol{v}\| \leq 1, \forall a \in \mathcal{A}, |\boldsymbol{u}^\top(\boldsymbol{\psi}_{\theta_{A(2)}}(a) \otimes \boldsymbol{v})| \leq M$.*

Note that unlike Xu et al. [35], we consider the case where the bridge function $h^*$ and the conditional expectation $\mathbb{E}_{W|A,Z}[\boldsymbol{\psi}_{\theta_W}]$ might not be included in the hypothesis spaces $\mathcal{H}_h$, $\mathcal{G}$, respectively.

As in Xu et al. [35], we leave aside questions of optimization. Thus, we assume that the optimization procedure over $(\theta_{A(1)}, \theta_Z, \boldsymbol{V})$ is sufficient to recover $\hat{\mathbb{E}}_{W|A,Z}[\boldsymbol{\psi}_{\theta_W}(W)]$, and that the optimization procedure over $(\theta_{A(2)}, \theta_W, \boldsymbol{u})$ is sufficient to recover $\hat{h}$ (which requires, in turn, the correct $\hat{\mathbb{E}}_{W|A,Z}$, for this $\boldsymbol{\psi}_{\theta_W}$). We emphasize that Algorithm 1 does not guarantee these properties. Based on these assumptions, we derive the generalization error in terms of $L_2$-norm with respect to $\mathbb{P}(A,W)$.

Following the discussion in Xu et al. [35, Lemma 1], we can show the following lemma that provides a generalization error bound for Stage 1 regression.

**Lemma 3.** *Under Assumption 7, and given stage 1 data $S_1 = \{(a_i, z_i, w_i)\}_{i=1}^{m}$, for any $\delta > 0$, with at least probability $1 - 2\delta$, we have*

$$\left\|\hat{\mathbb{E}}_{W|A,Z}\left[h(A,W)\right] - \mathbb{E}_{W|A,Z}\left[h(A,W)\right]\right\|_{P(A,Z)} \leq M\sqrt{\kappa_1 + 4\hat{\mathfrak{R}}_{S_1}(\mathcal{H}_1) + 24\sqrt{\frac{\log 2/\delta}{2m}}}$$

*for any $\boldsymbol{\psi}_{\theta_W} \in \mathcal{H}_{\boldsymbol{\psi}_{\theta_W}}$, where hypothesis space $\mathcal{H}_1$ is defined as*

$$\mathcal{H}_1 = \{(w,a,z) \in \mathcal{W} \times \mathcal{A} \times \mathcal{Z} \mapsto \|\boldsymbol{\psi}_{\theta_W}(w) - \boldsymbol{g}(a,z)\|^2 \in \mathbb{R} \mid \boldsymbol{g} \in \mathcal{G}, \boldsymbol{\psi}_{\theta_W} \in \mathcal{H}_{\boldsymbol{\psi}_{\theta_W}}\}, \quad (12)$$

*and $\kappa_1$ is the misspecificaiton error in the stage 1 regression defined as*

$$\kappa_1 = \max_{\boldsymbol{\psi}_{\theta_W} \in \mathcal{H}_{\boldsymbol{\psi}_{\theta_W}}} \min_{\boldsymbol{g} \in \mathcal{G}} \mathbb{E}_{W,A,Z}\left[\|\boldsymbol{\psi}_{\theta_W}(W) - \boldsymbol{g}(A,Z)\|^2\right]. \quad (13)$$

*Proof.* From Assumption 7, we have

$$\left\|\hat{\mathbb{E}}_{W|A,Z}\left[h(A,W)\right] - \mathbb{E}_{W|A,Z}\left[h(A,W)\right]\right\|_{P(A,Z)}$$

$$= \left\|\boldsymbol{u}^\top\left(\boldsymbol{\psi}_{\theta_{A(2)}}(a) \otimes \left(\mathbb{E}_{W|A,Z}\left[\boldsymbol{\psi}_{\theta_W}\right](a,z) - \hat{\mathbb{E}}_{W|A,Z}\left[\boldsymbol{\psi}_{\theta_W}\right](a,z)\right)\right)\right\|_{P(A,Z)}$$

$$= \sqrt{\mathbb{E}_{A,Z}\left[\left|\boldsymbol{u}^\top\left(\boldsymbol{\psi}_{\theta_{A(2)}}(a) \otimes \left(\mathbb{E}_{W|A,Z}\left[\boldsymbol{\psi}_{\theta_W}\right](a,z) - \hat{\mathbb{E}}_{W|A,Z}\left[\boldsymbol{\psi}_{\theta_W}\right](a,z)\right)\right)\right|^2\right]}$$

$$\leq \sqrt{\mathbb{E}_{A,Z}\left[M^2\left\|\mathbb{E}_{W|A,Z}\left[\boldsymbol{\psi}_{\theta_W}\right](a,z) - \hat{\mathbb{E}}_{W|A,Z}\left[\boldsymbol{\psi}_{\theta_W}\right](a,z)\right\|^2\right]}$$

$$= M\left\|\mathbb{E}_{W|A,Z}\left[\boldsymbol{\psi}_{\theta_W}\right](a,z) - \hat{\mathbb{E}}_{W|A,Z}\left[\boldsymbol{\psi}_{\theta_W}\right](a,z)\right\|_{P(A,Z)}$$

From Proposition 6, we have

$$\mathbb{E}_{W,A,Z}\left[\|\boldsymbol{\psi}_{\theta_W}(W) - \hat{\mathbb{E}}_{W|A,Z}[\boldsymbol{\psi}_{\theta_W}(W)](A,Z)\|^2\right]$$

$$\leq \hat{\mathcal{L}}_1(\hat{\mathbb{E}}_{W|A,Z}[\boldsymbol{\psi}_{\theta_W}(W)]) + 2\hat{\mathfrak{R}}_{S_1}(\mathcal{H}_1) + 12\sqrt{\frac{\log 2/\delta}{2m}}$$

with probability $1 - \delta$, since all functions $f \in \mathcal{H}_1$ satisfy $\|f\| \leq 4$ from $\|\boldsymbol{\psi}_{\theta_W}\| \leq 1, \|\boldsymbol{g}\| \leq 1$. Now, let $\boldsymbol{g}^*_{\theta_W}$ be

$$\boldsymbol{g}^*_{\theta_W} = \underset{\boldsymbol{g} \in \mathcal{G}}{\arg\min} \, \mathbb{E}_{W,A,Z}\left[\|\boldsymbol{\psi}_{\theta_W}(W) - \boldsymbol{g}(A,Z)\|^2\right].$$

Then, again from Proposition 6, we have

$$\hat{\mathcal{L}}_1(\boldsymbol{g}^*_{\theta_W}) \leq \mathbb{E}_{W,A,Z}\left[\|\boldsymbol{\psi}_{\theta_W}(W) - \boldsymbol{g}^*_{\theta_W}(A,Z)\|^2\right] + 2\hat{\mathfrak{R}}_{S_1}(\mathcal{H}_1) + 12\sqrt{\frac{\log 2/\delta}{2m}}$$

From the optimality of $\hat{\mathbb{E}}_{W|A,Z}[\boldsymbol{\psi}_{\theta_W}(W)]$, we have $\hat{\mathcal{L}}_1(\boldsymbol{g}^*_{\theta_W}) \geq \hat{\mathcal{L}}_1(\hat{\mathbb{E}}_{W|A,Z}[\boldsymbol{\psi}_{\theta_W}(W)])$, thus

$$\mathbb{E}_{W,A,Z}\left[\|\boldsymbol{\psi}_{\theta_W}(W) - \hat{\mathbb{E}}_{W|A,Z}[\boldsymbol{\psi}_{\theta_W}(W)](A,Z)\|^2\right]$$

$$\leq \mathbb{E}_{W,A,Z}\left[\|\boldsymbol{\psi}_{\theta_W}(W) - \boldsymbol{g}^*_{\theta_W}(A,Z)\|^2\right] + 4\hat{\mathfrak{R}}_{S_1}(\mathcal{H}_1) + 24\sqrt{\frac{\log 2/\delta}{2m}}. \quad (14)$$

holds with probability $1 - 2\delta$. Since we have

$$\mathbb{E}_{W,A,Z}\left[\|\boldsymbol{\psi}_{\theta_W}(W) - g(A,Z)\|^2\right]$$
$$= \mathbb{E}_{W,A,Z}\left[\|\boldsymbol{\psi}_{\theta_W}(W) - \mathbb{E}_{W|A,Z}[\boldsymbol{\psi}_{\theta_W}(W)]\|^2\right] + \mathbb{E}_{A,Z}\left[\|g(A,Z) - \mathbb{E}_{W|A,Z}[\boldsymbol{\psi}_{\theta_W}(W)]\|^2\right]$$

for all $\boldsymbol{g} \in \mathcal{G}$, by subtracting $\mathbb{E}_{W,A,Z}\left[\|\boldsymbol{\psi}_{\theta_W}(W) - \mathbb{E}_{W|A,Z}[\boldsymbol{\psi}_{\theta_W}(W)]\|^2\right]$ from both sides of (14), we have

$$\mathbb{E}_{A,Z}\left[\|\mathbb{E}_{W|A,Z}[\boldsymbol{\psi}_{\theta_W}(W)] - \hat{\mathbb{E}}_{W|A,Z}[\boldsymbol{\psi}_{\theta_W}(W)](A,Z)\|^2\right]$$

$$\leq \mathbb{E}_{A,Z}\left[\|\mathbb{E}_{W|A,Z}[\boldsymbol{\psi}_{\theta_W}(W)] - \boldsymbol{g}^*_{\theta_W}(A,Z)\|^2\right] + 4\hat{\mathfrak{R}}_{S_1}(\mathcal{H}_1) + 24\sqrt{\frac{\log 2/\delta}{2m}}$$

$$\leq \kappa_1 + 4\hat{\mathfrak{R}}_{S_1}(\mathcal{H}_1) + 24\sqrt{\frac{\log 2/\delta}{2m}}$$

By taking the square root of both sides, we have

$$\left\|\hat{\mathbb{E}}_{W|A,Z}[\boldsymbol{\psi}_{\theta_W}(W)] - \mathbb{E}_{W|A,Z}[\boldsymbol{\psi}_{\theta_W}(W)]\right\|_{P(A,Z)} \leq M\sqrt{\kappa_1 + 4\hat{\mathfrak{R}}_{S_1}(\mathcal{H}_1) + 24\sqrt{\frac{\log 2/\delta}{2m}}}$$

$$\square$$

The generalization error for Stage 2 can be shown by similar reasoning as in Xu et al. [35, Lemma 2].

**Lemma 4.** *Under Assumption 7, given stage 1 data $S_1 = \{(a_i, z_i, w_i)\}_{i=1}^m$, stage 2 data $S_2 = \{(\tilde{a}_i, \tilde{z}_i, \tilde{y}_i)\}_{i=1}^n$, and the estimated structural function $\hat{h}(a,z) = (\hat{\boldsymbol{u}})^\top (\boldsymbol{\psi}_{\hat{\theta}_{A(2)}}(a) \otimes \boldsymbol{\psi}_{\hat{\theta}_W}(w))$, then for any $\delta > 0$, with at least probability $1 - 4\delta$, we have*

$$\left\|\mathbb{E}_{Y|A,Z}[Y] - \hat{\mathbb{E}}_{W|A,Z}\left[\hat{h}(A,W)\right]\right\|_{P(A,Z)}$$

$$\leq \kappa_2 + M\sqrt{\kappa_1 + 4\hat{\mathfrak{R}}_{S_1}(\mathcal{H}_2) + 24\sqrt{\frac{\log 2/\delta}{2m}}} + \sqrt{4\hat{\mathfrak{R}}_{S_2}(\mathcal{H}_2) + 24M^2\sqrt{\frac{\log 2/\delta}{2n}}},$$

*where $\mathcal{H}_1$ is defined in (12), and $\mathcal{H}_2$ is defined as*

$$\mathcal{H}_2 = \{(y,a,z) \in \mathcal{Y} \times \mathcal{A} \times \mathcal{Z}$$
$$\mapsto (y - \boldsymbol{u}^\top(\boldsymbol{\psi}_{\theta_{A(2)}}(a) \otimes \boldsymbol{g}(a,z)))^2 \in \mathbb{R} \mid \boldsymbol{g} \in \mathcal{G}, \boldsymbol{u}, \theta_{A(2)}\}, \quad (15)$$

*and $\kappa_2$ is the misspecification error in Stage 2 defined as*

$$\kappa_2 = \min_{h \in \mathcal{H}_h} \left\|\mathbb{E}_{W|A,Z}[h(A,W)](A,Z) - \mathbb{E}_{Y|A,Z}[Y](A,Z)\right\|_{P(A,Z)}. \quad (16)$$

*Proof.* From Proposition 6, we have

$$\mathbb{E}_{Y,A,Z}\left[\left|Y - \hat{\mathbb{E}}_{W|A,Z}\left[\hat{h}(A,W)\right](A,Z)\right|^2\right] \leq \hat{\mathcal{L}}_2(\hat{h}) + 2\hat{\mathfrak{R}}_{S_2}(\mathcal{H}_2) + 12M^2\sqrt{\frac{\log 2/\delta}{2n}}$$

with probabiltiy $1 - \delta$, since all functions $f \in \mathcal{H}_2$ satisfy $\|f\| \leq 4M^2$ from $\|Y\| \leq M, |\boldsymbol{u}^\top(\boldsymbol{\psi}_{\theta_{A(2)}}(a) \otimes \boldsymbol{g})| \leq M$. Let $\tilde{h}$ be

$$\tilde{h} = \underset{h \in \mathcal{H}_h}{\arg\min} \left\|\mathbb{E}_{W|A,Z}[h(A,W)](A,Z) - \mathbb{E}_{Y|A,Z}[Y](A,Z)\right\|_{P(A,Z)}.$$

Again from Proposition 6, we have

$$\hat{\mathcal{L}}_2(\tilde{h}) \leq \mathbb{E}_{Y,A,Z}\left[\left|Y - \hat{\mathbb{E}}_{W|A,Z}\left[\tilde{h}(A,W)\right](A,Z)\right|^2\right] + 2\hat{\mathfrak{R}}_{S_2}(\mathcal{H}_2) + 12M^2\sqrt{\frac{\log 2/\delta}{2n}}.$$

with probabiltiy $1 - \delta$. From the optimality of $\hat{h}$, we have $\hat{\mathcal{L}}_2(\tilde{h}) \geq \hat{\mathcal{L}}_2(\hat{h})$, hence

$$\mathbb{E}_{Y,A,Z}\left[\left|Y - \hat{\mathbb{E}}_{W|A,Z}\left[\hat{h}(A,W)\right](A,Z)\right|^2\right]$$

$$\leq \mathbb{E}_{Y,A,Z}\left[\left|Y - \hat{\mathbb{E}}_{W|A,Z}\left[\tilde{h}(A,W)\right](A,Z)\right|^2\right] + 4\hat{\mathfrak{R}}_{S_2}(\mathcal{H}_2) + 24M^2\sqrt{\frac{\log 2/\delta}{2n}}$$

By subtracting $\mathbb{E}_{Y,A,Z}\left[\|Y - \mathbb{E}_{Y|A,Z}[Y]\|^2\right]$ from both sides, we have

$$\mathbb{E}_{A,Z}\left[\left|\mathbb{E}_{Y|A,Z}[Y] - \hat{\mathbb{E}}_{W|A,Z}\left[\hat{h}(A,W)\right](A,Z)\right|^2\right]$$

$$\leq \mathbb{E}_{A,Z}\left[\left|\mathbb{E}_{Y|A,Z}[Y] - \hat{\mathbb{E}}_{W|A,Z}\left[\tilde{h}(A,W)\right](A,Z)\right|^2\right] + 4\hat{\mathfrak{R}}_{S_2}(\mathcal{H}_2) + 24M^2\sqrt{\frac{\log 2/\delta}{2n}}$$

with probability $1 - 2\delta$. By taking the square root of both sides, with probability $1 - 2\delta$, we have

$$\|\mathbb{E}_{Y|A,Z}[Y] - \hat{\mathbb{E}}_{W|A,Z}\left[\tilde{h}(A,W)\right](A,Z)\|_{P(A,Z)}$$

$$\leq \sqrt{\mathbb{E}_{A,Z}\left[\left|\mathbb{E}_{Y|A,Z}[Y] - \hat{\mathbb{E}}_{W|A,Z}\left[\tilde{h}(A,W)\right](A,Z)\right|^2\right] + 4\hat{\mathfrak{R}}_{S_2}(\mathcal{H}_2) + 24M^2\sqrt{\frac{\log 2/\delta}{2n}}}$$

$$\leq \|\mathbb{E}_{Y|A,Z}[Y] - \hat{\mathbb{E}}_{W|A,Z}\left[\tilde{h}(A,W)\right](A,Z)\|_{P(A,Z)} + \sqrt{4\hat{\mathfrak{R}}_{S_2}(\mathcal{H}_2) + 24M^2\sqrt{\frac{\log 2/\delta}{2n}}}$$

$$\overset{(a)}{\leq} \left\|\mathbb{E}_{W|A,Z}\left[\tilde{h}(A,W)\right](A,Z) - \hat{\mathbb{E}}_{W|A,Z}\left[\tilde{h}(A,W)\right](A,Z)\right\|_{P(A,Z)}$$

$$+ \left\|\mathbb{E}_{W|A,Z}\left[\tilde{h}(A,W)\right](A,Z) - \mathbb{E}_{Y|A,Z}[Y](A,Z)\right\|_{P(A,Z)} + \sqrt{4\hat{\mathfrak{R}}_{S_2}(\mathcal{H}_2) + 24M^2\sqrt{\frac{\log 2/\delta}{2n}}}$$

$$\overset{(b)}{\leq} \kappa_2 + M\sqrt{\kappa_1 + 4\hat{\mathfrak{R}}_{S_1}(\mathcal{H}_2) + 24\sqrt{\frac{\log 2/\delta}{2m}}} + \sqrt{4\hat{\mathfrak{R}}_{S_2}(\mathcal{H}_2) + 24M^2\sqrt{\frac{\log 2/\delta}{2n}}}$$

where (a) holds from the triangular inequality and (b) holds from Lemma 3. □

Given the generalization errors in both stages, we can bound the error in (11) for the estimated bridge function $\hat{h}$. We need Assumption 3 to connect the error in (11) and the error $\|\hat{h} - h^*\|_{\mathbb{P}(A,W)}$. Let us restate Assumption 3.

**Assumption 3** (Completeness Assumption on Outcome-Inducing Proxy [3, 18]). *Let $l : \mathcal{W} \to \mathbb{R}$ be any square integrable function $\|l\|_{\mathbb{P}(W)} < \infty$, We assume the following condition:*

$$\mathbb{E}\left[l(W) \mid A = a, Z = z\right] = 0 \quad \forall(a,z) \in \mathcal{A} \times \mathcal{Z} \quad \Leftrightarrow \quad l(w) = 0 \quad \mathbb{P}(W)\text{-a.e.}$$

Given this assumption, we can consider the following constant $\tau_a$.

$$\tau_a = \underset{h \in L^2(\mathbb{P}(W|A=a)), h \neq h^*}{\max} \frac{\|h^*(a,W) - h(a,W)\|_{\mathbb{P}(W|A=a)}}{\|\mathbb{E}_{W|A=a,Z}[h^*(a,W)] - \mathbb{E}_{W|A=a,Z}[h(a,W)]\|_{\mathbb{P}(Z|A=a)}}$$

Note that Assumption 3 ensures $\tau_a < \infty$. We can bound the error using the supremum of this constant, $\tau = \sup_a \tau_a$.

**Theorem 2.** *Let Assumptions 3 and 7 hold. Given stage 1 data $S_1 = \{(w_i, a_i, z_i)\}_{i=1}^m$ and stage 2 data $S_2 = \{(\tilde{y}_i, \tilde{a}_i, \tilde{z}_i)\}_{i=1}^n$, for any $\delta > 0$, with at least probability of $1 - 6\delta$, we have*

$$\|h^*(A, W) - \hat{h}(A, W)\|_{P(A,W)}$$

$$\leq \tau \left( \kappa_2 + 2M\sqrt{\kappa_1 + 4\hat{\mathfrak{R}}_{S_1}(\mathcal{H}_1) + 24\sqrt{\frac{\log 2/\delta}{2m}}} + \sqrt{4\hat{\mathfrak{R}}_{S_2}(\mathcal{H}_2) + 24M^2\sqrt{\frac{\log 2/\delta}{2n}}} \right),$$

*where $\tau = \sup_a \tau_a$ and $\mathcal{H}_1, \kappa_1, \mathcal{H}_2, \kappa_2$ are defined in* (12), (13), (15), (16), *respectively.*

*Proof.*

$$\|h^*(A, W) - \hat{h}(A, W)\|_{P(A,W)}$$

$$\leq \sqrt{\mathbb{E}_A \left[ \|h^*(a, W) - \hat{h}(a, W)\|_{\mathbb{P}(W|A=a)}^2 \right]}$$

$$\leq \sqrt{\mathbb{E}_A \left[ \tau_a^2 \left\| \mathbb{E}_{W|A=a,Z} \left[ h^*(a, W) - \hat{h}(a, W) \right] \right\|_{\mathbb{P}(Z|A=a)}^2 \right]}$$

$$\leq \tau \sqrt{\mathbb{E}_A \left[ \left\| \mathbb{E}_{W|A=a,Z} \left[ h^*(a, W) - \hat{h}(a, W) \right] \right\|_{\mathbb{P}(Z|A=a)}^2 \right]}$$

$$\leq \tau \left\| \mathbb{E}_{Y|A,Z} [Y] - \mathbb{E}_{W|A,Z} \left[ \hat{h}(A, W) \right] \right\|_{\mathbb{P}(A,Z)}$$

$$\leq \tau \left\| \mathbb{E}_{Y|A,Z} [Y] - \hat{\mathbb{E}}_{W|A,Z} \left[ \hat{h}(A, W) \right] \right\|_{\mathbb{P}(A,Z)} + \tau \left\| \mathbb{E}_{W|A,Z} \left[ \hat{h}(A, W) \right] - \hat{\mathbb{E}}_{W|A,Z} \left[ \hat{h}(A, W) \right] \right\|_{\mathbb{P}(A,Z)}$$

Using Lemmas 3 and 4, the result thus follows. $\qquad\square$

From this result, we obtain the following corollary.

**Corollary 3.** *Let Assumption 7 hold and $\kappa_1, \kappa_2 = 0$. If $\hat{\mathfrak{R}}_{S_1}(\mathcal{H}_1) \to 0$ and $\hat{\mathfrak{R}}_{S_2}(\mathcal{H}_2) \to 0$ in probability as the dataset size increases, $\hat{h}$ converges to $h^*$ in probability with respect to $\|\cdot\|_{\mathbb{P}(A,W)}$.*

### C.3 Consistency Result for Causal Parameters

In this section, we develop a consistency result for causal parameters discussed in the main body. First, we consider the structural function. Given estimated bridge function $\hat{h}(a, w) = \hat{\boldsymbol{u}}^\top (\boldsymbol{\psi}_{\hat{\theta}_{A(2)}}(a) \otimes \boldsymbol{\psi}_{\hat{\theta}_W}(w))$, we estimate the structural function by taking the empirical mean over $W$. To make the discussion simple, we assume the access to an additional data sample $S_W = \{w_i^{\text{extra}}\}_{i=1}^{n_W}$, such that the estimated structural function is given as

$$\hat{f}_{\text{struct}}(a) = \hat{\boldsymbol{u}}^\top (\boldsymbol{\psi}_{\hat{\theta}_{A(2)}}(a) \otimes \boldsymbol{\mu}_{\hat{\theta}_W}), \tag{17}$$

where

$$\boldsymbol{\mu}_{\hat{\theta}_W} = \sum_{i=1}^{n_W} \boldsymbol{\psi}_{\hat{\theta}_W}(w_i^{\text{extra}}).$$

Note that empirically, we can use outcome-proxy data in $S_1$ instead of $S_W$.

To bound the deviation from the true structural function, we need the following assumption.

**Assumption 8.** *Let $\rho_W(w), \rho_{W|A}(w|a)$ be the respective density functions of probability distributions $\mathbb{P}(W), \mathbb{P}(W|A)$. The densities satisfy*

$$\eta_a := \left\| \frac{\rho_W(W)}{\rho_{W|A}(W|a)} \right\|_{\mathbb{P}(W|A=a)} < \infty.$$

Given Assumption 8, we can bound the error in structural function estimation. Before stating the theorem, let us introduce a useful concentration inequality for multi-dimensional random variables. [2]

**Lemma 5.** *Let $\boldsymbol{x}_1, \ldots, \boldsymbol{x}_n \in [-1, 1]^d$ be independent random variables. Then, with probability at least $1 - \delta$, we have*

$$\left\| \frac{1}{n} \sum_{i=1}^n \boldsymbol{x}_i - \boldsymbol{\mu} \right\| \leq \sqrt{\frac{2d \log 2d/\delta}{n}}$$

*where $\boldsymbol{\mu} = \mathbb{E}\left[\frac{1}{n} \sum_{i=1}^n \boldsymbol{x}_i\right]$.*

*Proof.* Let $j$-th coordinate of $\boldsymbol{x}_i$ be denoted as $(\boldsymbol{x}_i)_j$. Then,

$$\mathbb{P}\left( \left\| \frac{1}{n} \sum_{i=1}^n \boldsymbol{x}_i - \boldsymbol{\mu} \right\| \leq \varepsilon \right)$$

$$= \mathbb{P}\left( \sum_{j=1}^d \left( \frac{1}{n} \sum_{i=1}^n (\boldsymbol{x}_i)_j - (\boldsymbol{\mu})_j \right)^2 \leq \varepsilon^2 \right)$$

$$\leq \mathbb{P}\left( \bigcap_{j=1}^d \left| \frac{1}{n} \sum_{i=1}^n (\boldsymbol{x}_i)_j - (\boldsymbol{\mu})_j \right| \leq \frac{\varepsilon}{\sqrt{d}} \right)$$

$$\leq 1 - 2d \exp\left( -\frac{n\varepsilon^2}{2d} \right),$$

Where the last inequality holds from Hoeffding's inequality. The claim follows by solving $\delta = d \exp\left( -\frac{n\varepsilon^2}{2d} \right)$. $\square$

Given this lemma, we can prove the following result.

**Theorem 3.** *Let Assumptions 3, 7 and 8 hold. Let $\eta = \sup_{a \in \mathcal{A}} \eta_a$. Given stage 1 data $S_1 = \{(w_i, a_i, z_i)\}_{i=1}^m$, stage 2 data $S_2 = \{(\tilde{y}_i, \tilde{a}_i, \tilde{z}_i)\}_{i=1}^n$, additional outcome-proxy variable data $S_W = \{w_i^{\text{extra}}\}_{i=1}^{n_W}$, then with probability at least $1 - 7\delta$, we have*

$$\|f_{\text{struct}}^* - \hat{f}_{\text{struct}}\|_{\mathbb{P}(A)}$$

$$\leq \sqrt{\frac{2d_W \log 2d_W/\delta}{n_W}} + \eta\tau \left( \kappa_2 + 2M\sqrt{\kappa_1 + 4\hat{\mathfrak{R}}_{S_1}(\mathcal{H}_1) + 24\sqrt{\frac{\log 2/\delta}{2m}}} + \sqrt{4\hat{\mathfrak{R}}_{S_2}(\mathcal{H}_2) + 24M^2\sqrt{\frac{\log 2/\delta}{2n}}} \right),$$

*where $\hat{f}_{\text{struct}}$ is given in (17) and $d_W$ is the dimension of $\boldsymbol{\psi}_{\theta_W}$.*

*Proof.* From the relationship between structural function and bridge function, we have

$$\|f_{\text{struct}}^* - \hat{f}_{\text{struct}}\|_{\mathbb{P}(A)}$$

$$= \|\mathbb{E}_W[h^*(A, W)] - \hat{f}_{\text{struct}}\|_{\mathbb{P}(A)}$$

$$\leq \left\| \mathbb{E}_W[h^*(A, W)] - \mathbb{E}_W\left[\hat{h}(A, W)\right] \right\|_{\mathbb{P}(A)} + \left\| \mathbb{E}_W\left[\hat{h}(A, W)\right] - \hat{f}_{\text{struct}} \right\|_{\mathbb{P}(A)}.$$

---

[2]Lemma 5 is discussed in MathOverflow (https://mathoverflow.net/questions/186097/hoeffdings-inequality-for-vector-valued-random-variables, accessed on June 2nd 2021)

We can bound each term as follows. For the first term, we have

$$\left\|\mathbb{E}_W\left[h^*(\cdot, W)\right] - \mathbb{E}_W\left[\hat{h}(\cdot, W)\right]\right\|_{\mathbb{P}(A)}$$

$$= \sqrt{\int \left(\mathbb{E}_W\left[h^*(a, W) - \hat{h}(a, W)\right]\right)^2 \rho_A(a)\mathrm{d}a}$$

$$= \sqrt{\int \left(\mathbb{E}_{W|A=a}\left[(h^*(a, W) - \hat{h}(a, W))\frac{\rho_W(W)}{\rho_{W|A}(W|a)}\right]\right)^2 \rho_A(a)\mathrm{d}a}$$

$$\leq \sqrt{\int \left\|h^*(a, W) - \hat{h}(a, W)\right\|^2_{\mathbb{P}(W|A=a)} \left\|\frac{\rho_W(W)}{\rho_{W|A}(W|a)}\right\|^2_{\mathbb{P}(W|A=a)} \rho_A(a)\mathrm{d}a} \quad \because \text{Cauchy–Schwarz inequality}$$

$$\leq \sqrt{\int \eta_a^2 \mathbb{E}_{W|A=a}\left[(h^*(a, W) - \hat{h}(a, W))^2\right]\rho_A(a)\mathrm{d}a} \quad \because \text{Assumption 8}$$

$$\leq \eta\sqrt{\mathbb{E}_{W,A}\left[(h^*(A, W) - \hat{h}(A, W))^2\right]}$$

$$= \eta\|h^* - \hat{h}\|_{\mathbb{P}(A,W)}.$$

From Theorem 2, with probability at least $1 - 6\delta$, we have

$$\left\|\mathbb{E}_W\left[h^*(\cdot, W)\right] - \mathbb{E}_W\left[\hat{h}(\cdot, W)\right]\right\|_{\mathbb{P}(A)}$$

$$\leq \eta\tau\left(\kappa_2 + 2M\sqrt{\kappa_1 + 4\hat{\mathfrak{R}}_{S_1}(\mathcal{H}_1) + 24\sqrt{\frac{\log 2/\delta}{2m}}} + \sqrt{4\hat{\mathfrak{R}}_{S_2}(\mathcal{H}_2) + 24M^2\sqrt{\frac{\log 2/\delta}{2n}}}\right).$$

For the second term, from Assumption 7, we have with probability at least $1 - \delta$,

$$\left\|\mathbb{E}_W\left[\hat{h}(A, W)\right] - \hat{f}_{\text{struct}}\right\|_{\mathbb{P}(A)} \leq M\left\|\boldsymbol{\mu}_{\hat{\theta}_W} - \mathbb{E}_W\left[\boldsymbol{\psi}_{\hat{\theta}_W}\right]\right\|$$

$$\leq \sqrt{\frac{2d_W \log 2d_W/\delta}{n_W}}$$

The last inequality holds from Lemma 5. Using the uniform inequality, we have shown the claim. $\quad\square$

We can evaluate the error in estimating a value function as well. Given $S_3 = \{\breve{w}_i, \breve{c}_i\}_{i=1}^{n'}$, we estimate the value function as

$$\hat{v}(\pi) = \frac{1}{n'}\sum_{i=1}^{n}\hat{\boldsymbol{u}}^\top(\boldsymbol{\psi}_{\hat{\theta}_{A(2)}}(\pi(\breve{c}_i)) \otimes \boldsymbol{\psi}_{\hat{\theta}_W}(\breve{w}_i)),$$

Furthermore, we assume the following relationship between distributions of $A$ and $C$

**Assumption 9.** *There exists a constant $\sigma$ such that*

$$\|l(\pi(C), W)\|_{\mathbb{P}(C,W)} \leq \sigma\|l(A, W)\|_{\mathbb{P}(A,W)}$$

*for all square integrable functions $l : \mathcal{A} \times \mathcal{W} \to \mathbb{R}, \|l\| < \infty.$*

Given these assumptions, we have the following theorem.

**Theorem 4.** *Let Assumptions 3, 7, 8, 9 hold. Given stage 1 data $S_1 = \{(w_i, a_i, z_i)\}_{i=1}^{m}$, stage 2 data $S_2 = \{(\tilde{y}_i, \tilde{a}_i, \tilde{z}_i)\}_{i=1}^{n}$, stage 3 data $S_3 = \{\breve{w}_i, \breve{c}_i\}_{i=1}^{n'}$, with at least probability $1 - 7\delta$, we have*

$$|v(\pi) - \hat{v}(\pi)|$$

$$\leq \sigma\tau\left(\kappa_2 + 2M\sqrt{\kappa_1 + 4\hat{\mathfrak{R}}_{S_1}(\mathcal{H}_1) + 24\sqrt{\frac{\log 2/\delta}{2m}}} + \sqrt{4\hat{\mathfrak{R}}_{S_2}(\mathcal{H}_2) + 24M^2\sqrt{\frac{\log 2/\delta}{2n}}}\right)$$

$$+ \sqrt{\frac{M^2 \log 2/\delta}{n'}}.$$

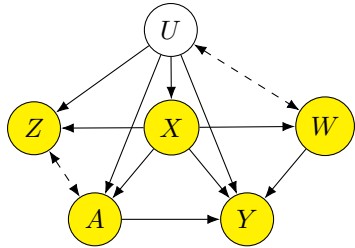

Figure 5: Causal graph with observable confounder

*Proof.* We have

$$|v(\pi) - \hat{v}(\pi)| = \left| \mathbb{E}_{W|C} \left[ h^*(\pi(C), W) \right] - \hat{v}(\pi) \right|$$

$$\leq \left| \mathbb{E}_{W,C} \left[ h(\pi(C), W) \right] - \mathbb{E}_{W,C} \left[ \hat{h}(\pi(C), W) \right] \right|$$

$$+ \left| \mathbb{E}_{W,C} \left[ \hat{h}(\pi(C), W) \right] - \hat{v}(\pi) \right|.$$

We bound each term as follows. For the first term, we have

$$\left| \mathbb{E}_{W,C} \left[ h^*(\pi(C), W) \right] - \mathbb{E}_{W,C} \left[ \hat{h}(\pi(C), W) \right] \right|$$

$$\leq \| h^*(\pi(C), W) - \hat{h}(\pi(C), W) \|_{\mathbb{P}(C,W)} \quad \because \text{Jensen's inequality}$$

$$\leq \sigma \| h^*(A, W) - \hat{h}(A, W) \|_{\mathbb{P}(A,W)} \quad \because \text{Assumption 9}$$

Hence, from Theorem 2, we have

$$\left| \mathbb{E}_{W,C} \left[ h^*(\pi(C), W) \right] - \mathbb{E}_{W,C} \left[ \hat{h}(\pi(C), W) \right] \right|$$

$$\leq \sigma\tau \left( \kappa_2 + 2M\sqrt{\kappa_1 + 4\hat{\mathfrak{R}}_{S_1}(\mathcal{H}_1) + 24\sqrt{\frac{\log 2/\delta}{2m}}} + \sqrt{4\hat{\mathfrak{R}}_{S_2}(\mathcal{H}_2) + 24M^2\sqrt{\frac{\log 2/\delta}{2n}}} \right)$$

with probability at least $1 - 6\delta$.

For the second term, we have

$$\left| \mathbb{E}_{W,C} \left[ \hat{\boldsymbol{u}}^\top \left( \boldsymbol{\psi}_{\hat{\theta}_{A(2)}}(\pi(C)) \otimes \boldsymbol{\psi}_{\hat{\theta}_W}(W) \right) \right] - \hat{v}(\pi) \right| \leq \sqrt{\frac{M^2 \log 2/\delta}{n'}}$$

with probability at least $1 - \delta$, from Hoeffding's inequality. The result is obtained by taking the union bound. $\qquad\square$

# D  DFPV algorithm with observable confounders

In this appendix, we formulate the DFPCL method when observable confounders are present, building on Tchetgen et al. [31] and Mastouri and Zhu et al. [18]. Here, we consider the causal graph given in Figure 5. In addition to variables $(A, Y, Z, W)$, we have an observable confounder $X \in \mathcal{X}$. The structural function $f_{\text{struct}}$ we aim to learn is

$$f_{\text{struct}}(a) = \mathbb{E}_{X,U} \left[ \mathbb{E} \left[ Y \mid A = a, X, U \right] \right].$$

The structural assumption and completeness assumption including observable confounders are given as follows.

**Assumption 10** (Structural Assumption [18]). *We assume $Y \perp\!\!\!\perp Z | A, U, X$, and $W \perp\!\!\!\perp (A, Z) | U, X$.*

**Assumption 11** (Completeness Assumption [18]). *Let $l : \mathcal{U} \to \mathbb{R}$ be any square integrable function $\|l\|_{\mathbb{P}(U)}$. We assume the following:*

$$\mathbb{E} \left[ l(U) \mid A = a, Z = z, X = x \right] = 0 \quad \forall(a, z, x) \in \mathcal{A} \times \mathcal{Z} \times \mathcal{X} \quad \Leftrightarrow \quad l(u) = 0 \text{ a.s.}$$

$$\mathbb{E} \left[ l(U) \mid A = a, Z = z, X = x \right] = 0 \quad \forall(a, z, x) \in \mathcal{A} \times \mathcal{Z} \times \mathcal{X} \quad \Leftrightarrow \quad l(u) = 0 \text{ a.s.}$$

Following similar reasoning as in Section 2, we can estimate the bridge function $\hat{h} : \mathcal{A} \times \mathcal{X} \times \mathcal{W} \to \mathbb{R}$ by minimizing the following loss:

$$\hat{h} = \underset{h \in \mathcal{H}_h}{\arg\min} \, \tilde{\mathcal{L}}(h), \quad \tilde{\mathcal{L}}(h) = \mathbb{E}_{Y,A,Z,X}\left[(Y - \mathbb{E}_{W|Z,A,X}[h(A,X,W)])^2\right] + \Omega(h).$$

Given bridge function, we can estimate the structural function by

$$f_{\text{struct}}(a) = \mathbb{E}_{X,W}\left[h^*(a,X,W)\right].$$

Similar to Mastouri and Zhu et al. [18], we model

$$\mathbb{E}_{W|a,z}[\boldsymbol{\psi}_{\theta_W}(w)] = \boldsymbol{V}(\boldsymbol{\phi}_{\theta_{A(1)}}(A) \otimes \boldsymbol{\phi}_{\theta_Z}(Z)) \otimes \boldsymbol{\phi}_{\theta_{X(1)}}(X))$$

$$h(a,x,w) = \boldsymbol{u}^\top(\boldsymbol{\psi}_{\theta_{A(2)}}(a) \otimes \boldsymbol{\psi}_{\theta_{X(2)}}(x) \otimes \boldsymbol{\psi}_{\theta_W}(w)),$$

where $\boldsymbol{\phi}_{\theta_{X(1)}}(X), \boldsymbol{\psi}_{\theta_{X(2)}}(X)$ are the feature maps of $X$ parameterized by $\theta_{X(1)}, \theta_{X(2)}$, respectively. Then, in stage 1, we learn $(\boldsymbol{V}, \theta_{A(1)}, \theta_Z, \theta_{X(1)})$ by minimizing

$$\hat{\mathcal{L}}_1(\boldsymbol{V}, \theta_{A(1)}, \theta_Z, \theta_{X(1)}) = \frac{1}{m}\sum_{i=1}^m \left\|\boldsymbol{\psi}_{\theta_W}(w_i) - \boldsymbol{V}\left(\boldsymbol{\phi}_{\theta_{A(1)}}(a_i) \otimes \boldsymbol{\phi}_{\theta_Z}(z_i) \otimes \boldsymbol{\phi}_{\theta_X}(x_i)\right)\right\|^2 + \lambda_1\|\boldsymbol{V}\|^2, \tag{18}$$

which estimates the conditional expectation $\mathbb{E}_W[\boldsymbol{\psi}_W(W)|A = a, Z = z, X = x]$. Let $(\hat{\boldsymbol{V}}, \hat{\theta}_{A(1)}, \hat{\theta}_Z, \hat{\theta}_{X(1)})$ be the minimizer of $\hat{\mathcal{L}}_1$. Then, in stage 2, we can learn $(\boldsymbol{u}, \theta_{A(2)}, \theta_Z, \theta_{X(2)})$ using

$$\hat{\mathcal{L}}_2(\boldsymbol{u}, \theta_{A(2)}, \theta_W, \theta_{X(2)}) = \frac{1}{n}\sum_{i=1}^n \left(\tilde{y}_i - \boldsymbol{u}^\top\left(\boldsymbol{\psi}_{\theta_{A(2)}}(\tilde{a}_i) \otimes \boldsymbol{\psi}_{\theta_{X(2)}}(\tilde{x}_i) \otimes \hat{\boldsymbol{V}}\boldsymbol{v}_1(\tilde{a}_i, \tilde{x}_i, \tilde{z}_i)\right)\right)^2 + \lambda_2\|\boldsymbol{u}\|^2, \tag{19}$$

where we denote $\boldsymbol{v}_1(a,x,z) = \left(\boldsymbol{\phi}_{\hat{\theta}_{A(1)}}(\tilde{a}_i) \otimes \boldsymbol{\phi}_{\hat{\theta}_X}(\tilde{x}_i) \otimes \boldsymbol{\phi}_{\hat{\theta}_Z}(\tilde{z}_i)\right)$.

In DFPV, we first fix parameters $\boldsymbol{\theta} = (\theta_{A(1)}, \theta_Z, \theta_{X(1)}, \theta_{A(2)}, \theta_Z, \theta_{X(2)})$ and obtain weights. This is given as

$$\hat{\boldsymbol{V}}(\boldsymbol{\theta}) = \Psi_1^\top \Phi_1 (\Phi_1^\top \Phi_1 + m\lambda_1 I)^{-1}, \qquad \hat{\boldsymbol{u}}(\boldsymbol{\theta}) = \left(\Phi_2^\top \Phi_2 + n\lambda_2 I\right)^{-1} \Phi_2^\top \boldsymbol{y}_2, \tag{20}$$

where we denote $\boldsymbol{\theta} = (\theta_{A(1)}, \theta_Z, \theta_{A(2)}, \theta_W)$ and define matrices as follows:

$$\Psi_1 = [\boldsymbol{\psi}_{\theta_W}(w_1), \dots, \boldsymbol{\psi}_{\theta_W}(w_m)]^\top, \qquad \Phi_1 = [\boldsymbol{v}_1(a_1, z_1), \dots, \boldsymbol{v}_1(a_m, z_m)]^\top,$$

$$\boldsymbol{y}_2 = [\tilde{y}_1, \dots, \tilde{y}_n]^\top, \qquad \Phi_2 = [\boldsymbol{v}_2(\tilde{a}_1, \tilde{z}_1), \dots, \boldsymbol{v}_2(\tilde{a}_n, \tilde{z}_n)]^\top,$$

$$\boldsymbol{v}_1(a,x,z) = \boldsymbol{\phi}_{\theta_{A(1)}}(a) \otimes \boldsymbol{\phi}_{\theta_{X(1)}}(x) \otimes \boldsymbol{\phi}_{\theta_Z}(z), \quad \boldsymbol{v}_2(a,z) = \boldsymbol{\psi}_{\theta_{A(2)}}(a) \otimes \boldsymbol{\psi}_{\theta_{X(2)}}(x) \otimes \left(\hat{\boldsymbol{V}}(\boldsymbol{\theta})\boldsymbol{v}_1(a,x,z)\right).$$

We learn the parameters by minimizing the following:

$$\hat{\mathcal{L}}_1^{\text{DFPV}}(\boldsymbol{\theta}) = \frac{1}{m}\sum_{i=1}^m \left\|\boldsymbol{\psi}_{\theta_W}(w_i) - \hat{\boldsymbol{V}}(\boldsymbol{\theta})\boldsymbol{v}_1(a_i, x_i, z_i)\right\|^2 + \lambda_1\|\hat{\boldsymbol{V}}(\boldsymbol{\theta})\|^2,$$

$$\hat{\mathcal{L}}_2^{\text{DFPV}}(\boldsymbol{\theta}) = \frac{1}{n}\sum_{i=1}^n \left(\tilde{y}_i - \hat{\boldsymbol{u}}(\boldsymbol{\theta})^\top \boldsymbol{v}_2(\tilde{a}_i, \tilde{x}_i, \tilde{z}_i)\right)^2 + \lambda_2\|\hat{\boldsymbol{u}}(\boldsymbol{\theta})\|^2.$$

The algorithm is given in Algorithn 2.

# E   Additional Experiments

In this appendix, we report the results of two additional experiments. One is a synthetic setting introduced in Mastouri et al. [18], which has a simpler data generating process. The other is based on the real-world setting introduced by Deaner [3]. In both setting, DFPV performs similarly to or better than existing methods.

**Algorithm 2** Deep Feature Instrumental Variable with Observable Confounder

---

**Input:** Stage 1 data $(a_i, z_i, w_i, x_i)$, Stage 2 data $(\tilde{a}_i \tilde{z}_i, \tilde{y}_i, \tilde{x}_i)$, Regularization parameters $(\lambda_1, \lambda_2)$. Initial values $\boldsymbol{\theta}^{(0)} = (\theta^{(0)}_{A(1)}, \theta^{(0)}_Z, \theta^{(0)}_{A(2)}, \theta^{(0)}_W, \theta^{(0)}_{X(1)}, \theta^{(0)}_{X(2)})$. Learning rate $\alpha$, additional data $(x^{\mathrm{extra}}_i, w^{\mathrm{extra}}_i)$

**Output:** Estimated structural function $\hat{f}_{\mathrm{struct}}(a)$

1: $t \leftarrow 0$
2: **repeat**
3:     Compute $\hat{V}(\boldsymbol{\theta}), \hat{u}(\boldsymbol{\theta})$ in (20)
4:     Update parameters in features $\boldsymbol{\theta}^{(t+1)} \leftarrow (\theta^{(t+1)}_{A(1)}, \theta^{(t+1)}_Z, \theta^{(t+1)}_{A(2)}, \theta^{(t+1)}_W, \theta^{(t+1)}_{X(1)}, \theta^{(t+1)}_{X(2)})$ by

$$\theta^{(t+1)}_{A(1)} \leftarrow \theta^{(t)}_{A(1)} - \alpha \nabla_{\theta_{A(1)}} \hat{\mathcal{L}}^{\mathrm{DFPV}}_1(\boldsymbol{\theta})|_{\boldsymbol{\theta}=\boldsymbol{\theta}^{(t)}}, \quad \theta^{(t+1)}_Z \leftarrow \theta^{(t)}_Z - \alpha \nabla_{\theta_Z} \hat{\mathcal{L}}^{\mathrm{DFPV}}_1(\boldsymbol{\theta})|_{\boldsymbol{\theta}=\boldsymbol{\theta}^{(t)}}$$

$$\theta^{(t+1)}_{X(1)} \leftarrow \theta^{(t)}_{X(1)} - \alpha \nabla_{\theta_{X(1)}} \hat{\mathcal{L}}^{\mathrm{DFPV}}_1(\boldsymbol{\theta})|_{\boldsymbol{\theta}=\boldsymbol{\theta}^{(t)}}, \quad \theta^{(t+1)}_{X(2)} \leftarrow \theta^{(t)}_{X(2)} - \alpha \nabla_{\theta_{X(2)}} \hat{\mathcal{L}}^{\mathrm{DFPV}}_2(\boldsymbol{\theta})|_{\boldsymbol{\theta}=\boldsymbol{\theta}^{(t)}}$$

$$\theta^{(t+1)}_{A(2)} \leftarrow \theta^{(t)}_{A(2)} - \alpha \nabla_{\theta_{A(2)}} \hat{\mathcal{L}}^{\mathrm{DFPV}}_2(\boldsymbol{\theta})|_{\boldsymbol{\theta}=\boldsymbol{\theta}^{(t)}}, \quad \theta^{(t+1)}_W \leftarrow \theta^{(t)}_W - \alpha \nabla_{\theta_W} \hat{\mathcal{L}}^{\mathrm{DFPV}}_2(\boldsymbol{\theta})|_{\boldsymbol{\theta}=\boldsymbol{\theta}^{(t)}}$$

5:     Increment counter $t \leftarrow t + 1$;
6: **until convergence**
7: Compute $\hat{u}(\bar{\boldsymbol{\theta}}^{(t)})$ from (20)
8: Compute mean feature for $W$ using stage 1 dataset

$$\boldsymbol{\mu}_{\theta_{X(2)} \otimes \theta_W} \leftarrow \frac{1}{n} \sum \boldsymbol{\psi}_{\theta^{(t)}_{X(2)}}(x^{\mathrm{extra}}_i) \otimes \boldsymbol{\psi}_{\theta^{(t)}_W}(w^{\mathrm{extra}}_i)$$

9: **return** $\hat{f}_{\mathrm{struct}}(a) = (\hat{u}^{(t)})^\top \left( \boldsymbol{\psi}_{\hat{\theta}^{(t)}_{A(2)}}(a) \otimes \boldsymbol{\mu}_{\theta_{X(2)} \otimes \theta_W} \right)$

---

## E.1 Experiments with Simpler Data Generating Process

Here, we show the result for the synthetic setting proposed in Mastouri et al. [18]. The data generating process for each variable is given as follows:

$$U := [U_1, U_2], \quad U_2 \sim \mathrm{Unif}[-1, 2] \quad U_1, \sim \mathrm{Unif}[0, 1] - \mathbb{1}[0 \leq U_2 \leq 1]$$
$$Z := [U_1 + \mathrm{Unif}[-1, 1], \, U_2 + \mathcal{N}(0, 3)]$$
$$W := [U_1 + \mathcal{N}(0, 3), \, U_2 + \mathrm{Unif}[-1, 1]]$$
$$A := U_2 + \mathcal{N}(0, 0.05)$$
$$Y := U_2 \cos(2(A + 0.3U_1 + 0.2))$$

From observations of $(Y, W, Z, A)$, we estimate $\hat{f}_{\mathrm{struct}}$ by PCL. For each estimated $\hat{f}_{\mathrm{struct}}$, we measure out-of-sample error as the mean square error of $\hat{f}$ versus true $f_{\mathrm{struct}}$ obtained from Monte-Carlo simulation. Specifically, we consider 20 evenly spaced values of $A \in [0.0, 1.0]$ as the test data. The results with data size $n = m = \{500, 1000\}$ are shown in Figure 6.

From Figure 6, we can see that DFPV and CEVAE methods perform worse and have larger variances than KPV and PMMR methods. This is not surprising, since DFPV tends to require more data than KPV and PMMR, as needed to learn the neural net feature maps (rather than using fixed pre-defined kernel features). Hence, we can say that we should favor KPV and PMMR over DFPV when the data is low-dimensional and the relations between the variables are smooth. We would like to note, however, DFPV outperforms CEVAE, which shows that the proxy setting is still required.

## E.2 Experiments using Grade Retention dataset

To test the performance of DFPV in a more realistic setting, we conducted the experiment on the Grade Retention dataset introduced by Deaner [3]. This aims to estimate the effect of grade retention based on the score of math and reading on the long-term cognitive outcomes, in which we use scores in elementary school as a treatment-inducing proxy (Z) and cognitive test scores from Kindergarten

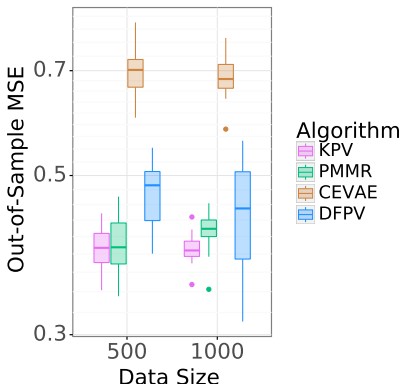

Figure 6: Result of structural function experiment in the setting in Mastouri et al. [18]

|  | DFPV | CEVAE | KPV | PMMR |
|---|---|---|---|---|
| Math | 0.023(0.001) | 0.054(0.007) | 0.043(0.000) | 0.032(0.001) |
| Reading | 0.027(0.002) | 0.082(0.007) | 0.028(0.000) | 0.022(0.000) |

Table 1: Results of grade retension dataset

as the an outcome-inducing proxy (W). Following Mastouri et al. [18], we generate a synthetic "ground truth" by fitting a generalized additive model to learn a structured causal model (SCM), and a Gaussian mixture model to learn unmeasured confounder based on the learned SCM. Note, this is needed since for real-world data there is no measured ground truth.

Table 1 shows the result for this dataset. In this setting, the performance of DFPV matches KPV and PMMR. As in the experiment described in the revious section, the setting is low-dimensional (one-dim treatment variable, three-dim treatment-inducing proxy, four-dim outcome-inducing proxy) and the generative model is smooth (the "ground truth" being a generalized additive model and a Gaussian mixture model). For these reasons, we might again expect this data to favor kernel methods, such as KPV and PMMR; nonetheless, our method matches them. DFPV again outperforms CEVAE in this setting.

## F  Experiment Details

In this section, we present the data generation process of experiments and the detailed settings of hyper-parameters.

### F.1  Demand Design Experiment

Here, we introduce the details of demand design experiments. The observations are generated from the following causal model,

$$Y = P\left(\exp\left(\frac{V-P}{10}\right) \wedge 5\right) - 5g(D) + \varepsilon, \quad \varepsilon \sim \mathcal{N}(0,1),$$

where $Y$ represents sales, $P$ is the treatment variable (price), and these are confounded by potential demand $D$. Here we denote $a \wedge b = \min(a,b)$, and the function $g$ as

$$g(d) = 2\left(\frac{(d-5)^4}{600} + \exp(-4(d-5)^2) + \frac{d}{10} - 2\right).$$

To correct this confounding bias, we introduce cost-shifter $C_1, C_2$ as a treatment-inducing proxy, and views $V$ of the reservation page as the outcome-inducing proxy. Data is sampled as

$$D \sim \text{Unif}[0, 10]$$
$$C_1 \sim 2\sin(2D\pi/10) + \varepsilon_1$$
$$C_2 \sim 2\sin(2D\pi/10) + \varepsilon_2$$
$$V \sim 7g(D) + 45 + \varepsilon_3$$
$$P = 35 + (C_1 + 3)h(D) + C_2 + \varepsilon_4$$

where $\varepsilon_1, \varepsilon_2, \varepsilon_3, \varepsilon_4 \sim \mathcal{N}(0, 1)$. From observations of $(Y, P, C_1, C_2, V)$, we estimate $\hat{f}_{\text{struct}}$ by PCL. For each estimated $\hat{f}_{\text{struct}}$, we measure out-of-sample error as the mean square error of $\hat{f}$ versus true $f_{\text{struct}}$ obtained from Monte-Carlo simulation. Specifically, we consider 10 evenly spaced values of $p \in [10, 30]$ as the test data.

### F.2 dSprite Experiment

Here, we describe the data generation process for the dSprites dataset experiment. This is an image dataset parametrized via five latent variables (`shape`, `scale`, `rotation`, `posX` and `posY`). The images are $64 \times 64 = 4096$-dimensional. In this experiment, we fixed the `shape` parameter to `heart`, i.e. we only used the heart-shaped images. The other latent parameters take values of `scale` $\in [0.5, 1]$, `rotation` $\in [0, 2\pi]$, `posX` $\in [0, 1]$, `posY` $\in [0, 1]$.

From this dataset, we generate the treatment variable $A$ and outcome $Y$ as follows:

1. Uniformly samples latent parameters $(\text{scale}, \text{rotation}, \text{posX}, \text{posY})$.

2. Generate treatment variable $A$ as
$$A = \text{Fig}(\text{scale}, \text{rotation}, \text{posX}, \text{posY}) + \boldsymbol{\eta}_A.$$

3. Generate outcome variable $Y$ as
$$Y = \frac{1}{12}(\text{posY} - 0.5)\frac{\|BA\|_2^2 - 5000}{1000} + \varepsilon, \quad \varepsilon \sim \mathcal{N}(0, 0.5).$$

Here, function `Fig` returns the corresponding image for the latent parameters, and $\boldsymbol{\eta}, \varepsilon$ are noise variables generated from $\boldsymbol{\eta}_A \sim \mathcal{N}(0.0, 0.1I)$ and $\varepsilon \sim \mathcal{N}(0.0, 0.5)$. Each element of the matrix $B \in \mathbb{R}^{10 \times 4096}$ is generated from $\text{Unif}(0.0, 1.0)$ and fixed throughout the experiment. From the data generation process, we can see that $A$ and $Y$ are confounded by `posY`. Treatment variable $A$ is given as a figure corrupted with Gaussian random noise. The variable `posY` is not revealed to the model, and there is no observable confounder. The structural function for this setting is

$$f_{\text{struct}}(A) = \frac{\|BA\|_2^2 - 5000}{1000}.$$

To correct this confounding bias, we set up the following PCL setting. We define the treatment-inducing variable $Z = (\text{scale}, \text{rotation}, \text{posX}) \in \mathbb{R}^3$, and the outcome-inducing variable by another figure that shares the same `posY`, with the remaining latent parameters fixed as follows:

$$W = \text{Fig}(0.8, 0, 0.5, \text{posY}) + \boldsymbol{\eta}_W,$$

where $\boldsymbol{\eta}_W \sim \mathcal{N}(0.0, 0.1I)$.

We use 588 test points for measuring out-of-sample error, which are generated from the grid points of latent variables. The grids consist of 7 evenly spaced values for `posX`, `posY`, 3 evenly spaced values for `scale`, and 4 evenly spaced values for `orientation`.

### F.3 Policy Evaluation Experiments

We use the same data $(Y, P, C_1, C_2, V)$ in demand design for policy evaluation experiments. We consider two policies. One is a policy depends on costs $C_1, C_2$ which is

$$\pi_{C_1, C_2}(C_1, C_2) = 23 + C_1 C_2.$$

To conduct offline-policy evaluation, we use data $(C_1, C_2, V)$ to compute the empirical average of $h(\pi_{C_1, C_2}(C_1, C_2), V)$. In our second experiment, the policy depends on current price $P$, which is

Table 2: Network structures of DFPV for demand design experiments. For the input layer, we provide the input variable. For the fully-connected layers (FC), we provide the input and output dimensions.

**Stage 1 Treatment Feature $\phi_{\theta_{A(1)}}$**

| Layer | Configuration |
|-------|---------------|
| 1 | Input($P$) |
| 2 | FC(1, 32), ReLU |
| 3 | FC(32, 16), ReLU |
| 4 | FC(16, 8), ReLU |

**Treatment-inducing Proxy Feature $\phi_{\theta_Z}$**

| Layer | Configuration |
|-------|---------------|
| 1 | Input($C_1, C_2$) |
| 2 | FC(2, 32), ReLU |
| 3 | FC(32, 16), ReLU |
| 4 | FC(16, 8), ReLU |

**Stage 2 Treatment Feature $\psi_{\theta_{A(2)}}$**

| Layer | Configuration |
|-------|---------------|
| 1 | Input($P$) |
| 2 | FC(1, 32), ReLU |
| 3 | FC(32, 16), ReLU |
| 4 | FC(16, 8), ReLU |

**Outcome-inducing Proxy Feature $\psi_{\theta_W}$**

| Layer | Configuration |
|-------|---------------|
| 1 | Input($V$) |
| 2 | FC(1, 32), ReLU |
| 3 | FC(32, 16), ReLU |
| 4 | FC(16, 8), ReLU |

given as
$$\pi_P(P) = \max(0.7P, 10).$$
Again, we use data $(P, V)$ to compute the empirical average of $h(\pi_P(P), V)$.

## F.4 Hyper-parameters and network architectures

Here, we describe the network architecture and hyper-parameters of all experiments.

For KPV and PMMR method, we used the Gaussian kernel where the bandwidth is determined by the median trick. We follow the procedure for hyper-parameter tuning proposed in Mastouri et al. [18] in selecting the regularizers $\lambda_1, \lambda_2$.

For DFPV, we optimize the model using Adam [10] with learning rate = 0.001, $\beta_1 = 0.9$, $\beta_2 = 0.999$ and $\varepsilon = 10^{-8}$. Regularizers $\lambda_1, \lambda_2$ are both set to 0.1 as a result of the tuning procedure described in Appendix A. Network structure is given in Tables 2, 3.

In CEVAE, we attempt to reconstruct the latent variable $L$ from $(A, Z, W, Y)$ using a VAE. Here, we use a 20-dim latent variable $L$, whose the conditional distribution is specified as follows:
$$q(L|A, Z, W, Y) = \mathcal{N}(\boldsymbol{V}_1\boldsymbol{\psi}_q(A, Z, W, Y), \mathrm{diag}(\boldsymbol{V}_2\boldsymbol{\psi}_q(A, Z, W, Y) \vee 0.1))$$
where $\boldsymbol{\psi}_q$ is a neural net and $\boldsymbol{V}_1, \boldsymbol{V}_2$ are matrices to be learned, and we denote $\boldsymbol{a} \vee 0.1 = (\max(a_i, 0.1))_i$. Furthermore, we specify the likelihood distribution as follows:
$$p(W, Z|L) = \mathcal{N}(\boldsymbol{V}_3\boldsymbol{\psi}_{p(W,Z|L)}(L), \mathrm{diag}(\boldsymbol{V}_4\boldsymbol{\psi}_{p(W,Z|L)}(L) \vee 0.1))$$
$$p(A|L) = \mathcal{N}(\boldsymbol{V}_5\boldsymbol{\psi}_{p(A|L)}(L), \mathrm{diag}(\boldsymbol{V}_6\boldsymbol{\psi}_{p(A|L))}(L) \vee 0.1))$$
$$p(Y|A, L) = \mathcal{N}(\boldsymbol{\mu}_{p(Y|A,L)}(A, L), 0.5)$$
Here, $\boldsymbol{\psi}_{p(W,Z|L)}, \boldsymbol{\psi}_{p(A|L)}, \boldsymbol{\mu}_{p(Y|A,L)}$ are neural networks. We provide the structure of neural nets in Table 4 and 5. Following the orignal work [17], we train all neural nets by Adamax [10] with a learning rate of 0.01, which was annealed with an exponential decay schedule. We also performed early stopping according to the lower bound on a validation set. To predict structural function, we obtain $q(L)$ by marginalizing $q(L|A, Z, W, Y)$ by observed data $A, Z, W, Y$. We then output $\hat{f}_{\mathrm{struct}}(a) = \mathbb{E}_{L \sim q(L)}\left[\mathbb{E}_{Y \sim p(Y|A=a, L)}[Y]\right]$.

Table 3: Network structures of DFPV for dSprite experiments. For the input layer, we provide the input variable. For the fully-connected layers (FC), we provide the input and output dimensions. SN denotes Spectral Normalization [23]. BN denotes Batch Normalization.

**Stage 1 Treatment Feature $\phi_{\theta_{A(1)}}$**

| Layer | Configuration |
|---|---|
| 1 | Input($A$) |
| 2 | FC(4096, 1024), SN, ReLU |
| 3 | FC(1024, 512), SN, ReLU, BN |
| 4 | FC(512, 128), SN, ReLU |
| 5 | FC(128, 32), SN, ReLU |

**Treatment-inducing Proxy Feature $\phi_{\theta_Z}$**

| Layer | Configuration |
|---|---|
| 1 | Input($Z$) |
| 2 | FC(3, 128), ReLU |
| 3 | FC(128, 64), ReLU |
| 4 | FC(64, 32), ReLU |

**Stage 2 Treatment Feature $\psi_{\theta_{A(2)}}$**

| Layer | Configuration |
|---|---|
| 1 | Input($A$) |
| 2 | FC(4096, 1024), SN, ReLU |
| 3 | FC(1024, 512), SN, ReLU, BN |
| 4 | FC(512, 128), SN, ReLU |
| 5 | FC(128, 32), SN, ReLU |

**Outcome-inducing Proxy Feature $\psi_{\theta_W}$**

| Layer | Configuration |
|---|---|
| 1 | Input($W$) |
| 2 | FC(4096, 1024), SN, ReLU |
| 3 | FC(1024, 512), SN, ReLU, BN |
| 4 | FC(512, 128), SN, ReLU |
| 5 | FC(128, 32), SN, ReLU |

Table 4: Network structures of CEVAE for demand design experiment. For the input layer, we provide the input variable. For the fully-connected layers (FC), we provide the input and output dimensions.

**Structure of $\psi_q$**

| Layer | Configuration |
|---|---|
| 1 | Input($P, Y, C_1, C_2, V$) |
| 2 | FC(5, 128), ReLU |
| 3 | FC(128, 64), ReLU |
| 4 | FC(64, 32), ReLU |

**Structure of $\psi_{p(W,Z|L)}$**

| Layer | Configuration |
|---|---|
| 1 | Input($L$) |
| 2 | FC(20, 64), ReLU |
| 3 | FC(64, 32), ReLU |
| 4 | FC(32, 16), ReLU |

**Structure of $\psi_{p(A|L)}$**

| Layer | Configuration |
|---|---|
| 1 | Input($L$) |
| 2 | FC(20, 64), ReLU |
| 3 | FC(64, 32), ReLU |
| 4 | FC(32, 16), ReLU |

**Structure of $\mu_{p(Y|A,L)}$**

| Layer | Configuration |
|---|---|
| 1 | Input($L, P$) |
| 2 | FC(21, 64), ReLU |
| 3 | FC(64, 32), ReLU |
| 4 | FC(32, 16), ReLU |
| 5 | FC(16, 1) |

Table 5: Network structures of CEVAE for dSprite experiment. For the input layer, we provide the input variable. For the fully-connected layers (FC), we provide the input and output dimensions. SN denotes Spectral Normalization [23]. BN denotes Batch Normalization.

**Structure of $\psi_q$**

| Layer | Configuration |
|-------|---------------|
| 1 | Input($W, A, Z, Y$) |
| 2 | FC(8196, 1024), SN, ReLU |
| 3 | FC(1024, 512), SN, ReLU, BN |
| 4 | FC(512, 128), SN, ReLU |
| 5 | FC(128, 32), SN, ReLU |

**Structure of $\psi_{p(W,Z|L)}$**

| Layer | Configuration |
|-------|---------------|
| 1 | Input($L$) |
| 2 | FC(20, 64), ReLU |
| 3 | FC(64, 128), ReLU |
| 4 | FC(128, 256), ReLU |

**Structure of $\psi_{p(A|L)}$**

| Layer | Configuration |
|-------|---------------|
| 1 | Input($L$) |
| 2 | FC(20, 64), ReLU |
| 3 | FC(64, 128), ReLU |
| 4 | FC(128, 256), ReLU |

**Structure of $\mu_{p(Y|A,L)}$**

| Layer | Configuration |
|-------|---------------|
| 1 | Input($L, A$) |
| 2 | FC(4116, 1024), SN, ReLU, BN |
| 3 | FC(1024, 512), SN, ReLU, BN |
| 4 | FC(512, 128), SN, ReLU |
| 5 | FC(128, 32), SN, ReLU |
| 6 | FC(32, 1) |