# OpenReview forum: "Deep Proxy Causal Learning and its Application to Confounded Bandit Policy Evaluation"
_NeurIPS.cc/2021/Conference — NeurIPS 2021 Poster_

### Official Review · Reviewer_Rp98 · 2021-07-09

**Rating:** 7
**Confidence:** 3

**Summary:**

This work extends known Proxy Causal Learning methods that aim at learning the causal effect of treatment in cases where unobserved confounders are assumed yet there is observable information in form of proxy variables to these confounders.
In particular, authors extend the theory to allow deep neural nets to be used as building blocks of the two stage regression proposed by Mastouri et al instead of RKHS. The paper also explains how to use the method to perform off-policy evaluation in a bandit setting where arms are confounded. Synthetic experiments are also reported to compare the performance with a few baselines of the same family.

**Limitations And Societal Impact:**

I would be keen to read a developed argument about some of the claims in the discussion phase. I believe it could firm up my mind on the significance of the proposed method.

- l32: authors indicate that ref 7/9/33 (BART, CFRNet, ...) have too restrictive assumptions; in what sense exactly ? if they are too restrictive can you give a small example explaining the limitations ?
- l42: authors indicate that ref 14/17 provide "little theory" to guarantee recovery of causal effect - could you elaborate and point out what is missing in your opinion and how your work differs ?
- l60: authors seem to imply that deepnets can learn more complex functions than RKHS - is that right ? if so in what sense ? are the cases where it makes a difference significant from an application perspective ?
- l101: the proposed setup assumes the existence of both outcome- and treatment-inducing proxies - isn't that a bit restrictive ? esp. compared to latent variables methods like the Deconfounder or ref 14/17 ?
- l207: did you try to measure such Rademacher complexities to compute this bound in practice (and verify its empirical tightness) ?
- in Sec 4, it seems to me the proposed setups are maybe more complex than it should. Esp. the Demand experiment equations seem a bit random and not very realistic (just try to imagine the shape of the curve from the equations in your head ?). At this point I can't make up my mind if the setup was designed to be so arbitrary that deepnets would be the only viable modeling option or if it resembles anything realistic.
- in the same idea I'd like to see what the method does when the setup is simpler. I believe it is part of the authors duty to study the limits of their method and the regimes where it is recommended to use it or not.
- the choice of CEVAE as the only baseline not from the CPL family is not very well motivated and I wonder of other methods could have been tried as well ?
- Ideally, I would like to have pointers on the performance of these baselines in the same setup in different papers to make sure the baseline methods perform as expected (i.e. why another benchmark ?)

**Main Review:**

Originality:
- the two main original contributions seem to be the theory that enables deepnets in place of RKHS and the explanation of how to do OPE with the method

Quality & Clarity:
- I appreciate the summary of related works which makes it easy to refresh one's memory before diving in the main development
- The notation and mathematical developments are relatively easy to follow thanks to a nice introduction of details and assumptions when they are needed
- I appreciate also that the corresponding proofs in appendix are detailed enough to follow and that required a dedicated effort from the authors
- I'm not completely convinced by some of the arguments, especially on why we need deepnets (see below)
- I'm not convinced that the experimental setup is providing evidence for that either (see below)
- there are a few typos (l242 Propositionn) but it doesn't affect reading

Significance:
- I understand in principle that the work enables application of PCL to more complex settings, even though I think the authors don't make an especially good argument on why it is required (the running argument in my head being that it might be the next natural thing to do in the field if deepnets were not tried before)
- the OPE reduction is usable for all PCL methods and thus might be interesting for this community which means for a large number of potential application domains where off-policy is interesting or the only option possible


**Time Spent Reviewing:**

4

---

> ### Author Response · Authors · 2021-08-10
> **Response to Reviewer Rp98**
>
> Thank you for the thoughtful questions. Clarifying these would definitely improve our paper, and we would like to incorporate the feedback in our revision. Please check if our responses below are satisfying.
>
> ---
> **Question 1:**
> > l32: authors indicate that ref 7/9/33 (BART, CFRNet, ...) have too restrictive assumptions; in what sense exactly? if they are too restrictive can you give a small example explaining the limitations?
>
>  **Answer:** These works make the "ignorable treatment assignment" assumption, which essentially means there are no unobservable confounders. However, this is rarely satisfied in real-world applications. Consider the problem of estimating the impact of smoking on life expectancy. There are many possible confounders, such as income or exercise habits, which are difficult to measure for both technical and ethical reasons. In such a case, we cannot use methods reliant on observation of the confounders.
>
> **Question 2:**
> > l42: authors indicate that ref 14/17 provide "little theory" to guarantee recovery of causal effect - could you elaborate and point out what is missing in your opinion and how your work differs?
>
> **Answer:** Ref 14/17 propose to recover an unobserved confounder based on a proxy variable. Such recovery, however, is not guaranteed to be successful, and these works did not describe the formal conditions and assumptions that enable us to learn a correct structural function. Our work has strong theoretical guarantees and can provably estimate the true causal effect given a sufficient amount of data, and under the conditions stated.
>
> **Question 3:**
> > l60: authors seem to imply that deepnets can learn more complex functions than RKHS - is that right? if so in what sense ? are the cases where it makes a difference significant from an application perspective?
>
>  **Answer:** Although there are ongoing research efforts to justify the superiority of deep networks over RKHS functions (or linear estimators, more generally), empirically, deep networks work better than kernel methods when the data is structured and high-dimensional (e.g. images/text). Theoretically, deep learning is known to be superior to linear estimators including kernel ridge regression, in estimating functions with spatially inhomogeneous smoothness  (functions in Besov spaces) (Suzuki, 2019), where adaptive features are effective.
>
> **Question 4:**
> > l101: the proposed setup assumes the existence of both outcome- and treatment-inducing proxies - isn't that a bit restrictive? esp. compared to latent variables methods like the Deconfounder or ref 14/17 ?
>
> **Answer:** We agree that having two types of proxy variables can be restrictive, but this is a necessary condition for learning the true causal effect. (See ref 13 for details). Latent variable methods seem to require less restrictive conditions, but the learned causal effect is not guaranteed to converge to the true causal effect in general.
>
> **Question 5:**
> > l207: did you try to measure such Rademacher complexities to compute this bound in practice (and verify its empirical tightness)?
>
>  **Answer:** We note that Rademacher complexity is impossible to compute in practice, since we have to optimize a model for each set of Rademacher variables, which requires us to solve $2^n$ optimization problems. It may yet vanish, but this is a challenging open question: see the reply to Reviewer bAic.
>
> **Question 6:**
> > in Sec 4, it seems to me the proposed setups are maybe more complex than it should. Esp. the Demand experiment equations seem a bit random and not very realistic (just try to imagine the shape of the curve from the equations in your head ?). At this point I can't make up my mind if the setup was designed to be so arbitrary that deepnets would be the only viable modeling option or if it resembles anything realistic.
>
>  **Answer:** We agree that the Demand setting is synthetic, but this benchmark is designed to test the ability to learn complex bridge functions. To illustrate the performance in a realistic setting (albeit low dimensional), we will include the experiment of estimating the causal effect of grade retention proposed in (ref. 3). See "official comment to all reviewers."
>
> **Questions 7 and 8:**
> > in the same idea I'd like to see what the method does when the setup is simpler. I believe it is part of the authors duty to study the limits of their method and the regimes where it is recommended to use it or not.
> the choice of CEVAE as the only baseline not from the CPL family is not very well motivated and I wonder of other methods could have been tried as well?
>
> > Ideally, I would like to have pointers on the performance of these baselines in the same setup in different papers to make sure the baseline methods perform as expected (i.e. why another benchmark ?)
>
> **Answer:** Thank you for the suggestion. We will include the experimental result for the simpler setting proposed in (ref 18). Please refer to the overall official comment for details. We would like to include other baselines, but as far as we know, we are not aware of other methods that we can try (other than ref. 14, which replace the VAE model in CEVAE with a  Generative Adversarial Networks, but does not otherwise address the deeper modelling assumptions of that approach).
>
> ### Reference:
> * Taiji Suzuki: Adaptivity of deep ReLU network for learning in Besov and mixed smooth Besov spaces: optimal rate and curse of dimensionality. The 7th International Conference on Learning Representations (ICLR2019)

---

> > ### Comment · Reviewer_Rp98 · 2021-09-02
> > **thank you for your answers**
> >
> > esp. on the difference with ignorability settings.
> > I've updated my score accordingly.

---

### Official Review · Reviewer_q48d · 2021-07-16

**Rating:** 6
**Confidence:** 4

**Summary:**

The paper proposes an algorithm that operationalize the proxy variable
strategy for causal identification. It builds on existing approaches
(Deaner [3] and Mastouri and Zhu et al. [18]) and additionally
consider adaptive features as in Deep Feature Instruemtal Variable
method [32]. The key idea is to consider features parameterized by
neural nets, and alternate between learning $V$ by minimizing the
empirical loss and learning the parameters of the feature maps. The
paper shows that the additional flexibility of the features helps the
proxy variable method achieve better out of sample performance and
smaller errors in empirical studies.

**Limitations And Societal Impact:**

The paper could benefit from a more elaborate discussion on its limitations. Is the use of adaptive features free lunch?

**Main Review:**

The paper proposes an algorithm that operationalize the proxy variable
strategy for causal identification. It builds on existing approaches
(Deaner [3] and Mastouri and Zhu et al. [18]) and additionally
consider adaptive features as in Deep Feature Instruemtal Variable
method [32]. The key idea is to consider features parameterized by
neural nets, and alternate between learning $V$ by minimizing the
empirical loss and learning the parameters of the feature maps. The
paper shows that the additional flexibility of the features helps the
proxy variable method achieve better out of sample performance and
smaller errors in empirical studies.

Both the theoretical and empirical results of the paper make sense.
The methods and the results of the paper build closely on the two
stage formulation of Deaner [3] and Mastouri and Zhu et al. [18] and
the adaptive feature idea in Deep Feature Instruemtal Variable method
[32]. I have a few major questions.

The paper claim that Assumption 1 and 2 (the completeness assumption
on the unobserved confounder) are sufficient for causal
identification. In particular, the paper says that: Assumption 3,
which was needed in the original proxy variable paper (Miao et al
2018), is not necessary. Can the authors explain the intuition here?
Why is it not necesary here? The proof of Miao et al 2018 does rely on
this assumption. Moreover, do these completeness conditions in
Assumptions 1 and 2 have observable implications? Do they imply the
conditional of the proxy given treatment is complete in any ways? Or
do they have provably *no* observable implications? These observables
implications are important for causal estimation in principle.

Further, the completeness assumptions  plays a key role for
identification in the proxy variable strategy. They are not regularity
conditions. Rather, the completeness conditions are quite strong; they
are used to guarantee the solution to the integral equation is unique.
(Solutions to integral equations are not unique in general.) Despite
the importance of these identification assumptions, nowhere in the
algorithm are these completeness conditions enforced. As above, even
though the completeness assumption of Assumptions 1 and 2 are about
unobserved confounders, they may have observable implications and
shall be enforced in the proposed algorithm. (Or we should prove that
there exist *no* observable implications.) Without enforcing these
conditions, the causal estimation can in general by biased.

Finally, the paper could benefit from a discussion on the connections
to other recent work on proxy variables, which also discussed
practical estimation approaches to proxy variables that may be
compatible with the adaptive feature idea:

Miao, W., Hu, W., Ogburn, E. L., & Zhou, X. (2020). Identifying effects of multiple treatments in the presence of unmeasured confounding. arXiv preprint arXiv:2011.04504.


Wang, Y., & Blei, D. (2021, July). A Proxy Variable View of Shared Confounding. In International Conference on Machine Learning (pp. 10697-10707). PMLR.

Miao, W., & Tchetgen Tchetgen, E. (2018). A confounding bridge approach for double negative control inference on causal effects (supplement and sample codes are included). arXiv preprint arXiv:1808.04945.

**Time Spent Reviewing:**

10

---

> ### Author Response · Authors · 2021-08-10
> **Response to Reviewer q48d**
>
> Thank you for the insightful questions. Our responses to them are as follows.
>
> ---
> **Question 1:**
>  > The paper claim that Assumption 1 and 2 (the completeness assumption on the unobserved confounder) are sufficient for causal identification. In particular, the paper says that: Assumption 3, which was needed in the original proxy variable paper (Miao et al 2018), is not necessary. Can the authors explain the intuition here? Why is it not necesary here? The proof of Miao et al 2018 does rely on this assumption.
>
> **Answer:** We would like to point out that Proposition 1 is a relatively weak claim --- it only indicates when a bridge function exists.
> As we argue in Section B.1, Assumptions 1 and 2 are sufficient for the existence claim, which is independent of bridge function estimators.
> Note that (the expectation of) any bridge function yields the structural function, and the uniqueness is therefore not necessary.  On the other hand, the stronger assumption (Assumption 3) is used to guarantee the consistent estimation of causal quantities (Proposition 2 and 3) with our two-stage regression estimation; therefore, our work effectively follows (Miao et al., 2018) in terms of the required assumptions.
>
> Regarding the difference between Assumptions 2 and 3 in proving Proposition 1, the latter is a stronger condition to establish that the target $\mathbb{E}[Y|A=a, Z=\cdot]$ is in the range of the integral operator $E_a$ of the required integral equation; the first completeness condition in Assumption 2 is weaker and suffices for this purpose (please see Remark 2 in Appendix B.1).
>
> **Question 2:**
> > Moreover, do these completeness conditions in Assumptions 1 and 2 have observable implications? Do they imply the conditional of the proxy given treatment is complete in any ways? Or do they have provably no observable implications? These observables implications are important for causal estimation in principle.
>     Further, the completeness assumptions plays a key role for identification in the proxy variable strategy. They are not regularity conditions. Rather, the completeness conditions are quite strong; they are used to guarantee the solution to the integral equation is unique. (Solutions to integral equations are not unique in general.) Despite the importance of these identification assumptions, nowhere in the algorithm are these completeness conditions enforced. As above, even though the completeness assumption of Assumptions 1 and 2 are about unobserved confounders, they may have observable implications and shall be enforced in the proposed algorithm. (Or we should prove that there exist no observable implications.) Without enforcing these conditions, the causal estimation can in general by biased.
>
> **Answer:** The completeness assumptions are distributional assumptions about observed data. Although sufficient conditions for $L^2$-completeness are known under specific circumstances (e.g., exponential family distributions (Newey and Powell, 2003); see (Andrews, 2017) for a review), to our knowledge, there are no known observable implications. As such, it is challenging to enforce a completeness assumption in algorithms like ours. We admit that the completeness assumptions are strong and difficult to validate in practice; relaxing the assumptions (or verifying them) is certainly desirable, and our work leaves open this possibility (as is the case for previous work in the area, [3, 18, 20]). While there is room for theoretical improvement, some machine learning applications may still benefit from the proposed method, such as reinforcement learning [30] or confounded bandit problems, as we demonstrated in our experiments.
>
> **Question 3:**
> > Finally, the paper could benefit from a discussion on the connections to other recent work on proxy variables, which also discussed practical estimation approaches to proxy variables that may be compatible with the adaptive feature idea:
>
> **Answer:** We appreciate the suggested references. We will include them and discuss the relation to them. In brief, the first paper develops another method of causal inference based on auxiliary variables, which relies on different grounding assumptions to Proxy Causal Learning. The second paper interprets Wang \& Blei (2019) in the setting of Proxy Causal Learning, and addresses the case of a large number of proxies. We actually included the different version of the third paper, in [21], which presents the sufficient condition of existence of the bridge function. We will update the reference for this.
>
> **Question 4:**
> > The paper could benefit from a more elaborate discussion on its limitations. Is the use of adaptive features free lunch?
>
> **Answer:** Unfortunately, the use of adaptive features is no free lunch. As with other deep learning methods, DFPV typically requires more data than kernel-based methods. We will include a discussion of this limitation, and demonstrate it with an additional experimental result, with a smaller data size and a simpler data generation process (as proposed in [18]), for which DFPV performs slightly worse than KPV and PMMR. See the "official comment to all reviewers" for details.
>
> ---
> ### References:
> * Newey, W., & Powell, J. (2003). Instrumental Variable Estimation of Nonparametric Models. Econometrica, 71(5), 1565-1578.
> * Andrews, D. W. (2017): “Examples of l2-complete and boundedly-complete distributions,” Journal of Econometrics, 199(2), 213–220.

---

> > ### Comment · Reviewer_q48d · 2021-08-31
> > **Thank you for your response**
> >
> > Thank you for your response. My (positive) evaluation of the paper stays.

---

### Official Review · Reviewer_fc4M · 2021-07-16

**Rating:** 6
**Confidence:** 3

**Summary:**

This paper presents a novel method: deep feature proxy variable. It applies a neural network to PCL, allowing non-linear relationships between variables. The authors prove the consistency of the method and demonstrate how this method can be applied to off-policy evaluation. The authors test the methods using two simulation studies, showing that DFPV outperforms existing methods.

**Limitations And Societal Impact:**

The authors addressed the limitation of the theoretical performance, showing that the method works well in practice. I would like to see an non-trivial failure case of the method.

**Main Review:**

Overall, I enjoyed reading this paper. I believe some analysis on the usability of the method could make the paper more impactful. I would a consider increasing my score if the authors address the concerns.

pros
- The paper is clear and well written. It’s clear in its notation and setup. It also clearly stated the limitations of the theoretical contribution.
- The paper is well researched and motivated. It provides a crisp overview of proximal causal learning. It articulates the challenges in PCL and provides a reasonable solution.
- The experiments are reasonably designed. The authors provide detailed records of the experiment and source code.

Area of improvements
- The captions in figure 3 and 4 are not clear. Why is DFPV so high variance?
-  I would be interested in knowing the performance of DFPV on a relatively simple data generating process. What sort of behaviour would DFPV exhibit in that case?
- I would like an analysis about when we should use DFPV in contrast to the parametric methods.


**Time Spent Reviewing:**

2

---

> ### Author Response · Authors · 2021-08-10
> **Response to Reviewer fc4M**
>
> Thank you for the positive comment! Please check our responses to your concerns below.
>
> ---
> **Question 1:**
> > The captions in figure 3 and 4 are not clear. Why is DFPV so high variance?
>
> **Answer:** We would like to note that the variance of DFPV is not higher than other methods. It seems wider in the figure since we are using logarithmic y-axis.
>
> **Question 2:**
> > I would be interested in knowing the performance of DFPV on a relatively simple data generating process. What sort of behaviour would DFPV exhibit in that case?
>
> **Answer:** Thank you for the suggestion. In response to the comment, we will include the experiment with a relatively simple data generating process, in which DFPV performs slightly worse than KPV or PMMR. Please refer to the overall comments for more detail.
>
> **Question 3:**
> > I would like an analysis about when we should use DFPV in contrast to the parametric methods.
>
> **Answer:** We should use DFPV when treatment/proxy variables are high-dimensional and structured (e.g. images or text), and/or when the bridge function is very complex. This is supported by the experimental results. The aim of the Demand task is to test whether the methods can learn a complex bridge function, and the aim of the dSprite task is to see whether they can handle high-dimensional proxy variables. We show that DFPV is superior to other methods in both cases.

---

### Official Review · Reviewer_bAic · 2021-08-01

**Rating:** 7
**Confidence:** 3

**Summary:**

The work proposes a causal effect estimation method in presence of unobserved confounding, building on proximal causal learning. It starts from an existing two-stage regression approach for tractably solving the proximal causal learning problem and extends it to handle high-dimensional and non-linearly related data using neural networks to learn adaptive features which are used as input to the regression approach.

**Ethical Concerns:**

I do not foresee any ethical implications of the presented applications. As a positive implication, the work enables making more robust causal inferences when unobserved confounders exist. Thus, it may be applicable to problems in social sciences where all possible confounders cannot be measured such as socio economic indicators of individuals in treatment efficacy studies.

**Limitations And Societal Impact:**

Consider adding limitation on only handling continuous actions.

Suggestions:

Average treatment *effect* is defined typically as the contrast between potential outcomes at two treatment levels and not for a single treatment level as in line 88. Consider rewording to average potential outcome or a similar name.

The experiment setup for demand design experiment states it as a prediction task line 277. Instead, please consider posing it as a causal effect estimation task to justify the need for removing confounding bias.

Assumptions 4,5,6 directly referenced in the Proposition 1 without introducing the nature of these assumptions. Please add a few words e.g. saying these are regularity conditions on conditional expectation functions or that these are not structural but functional assumptions.
Similarly, for Assumptions 7,8.

The approximation for computing gradient w.r.t. theta_W as mentioned in line 171 can be clarified in the update equations in Algorithm 1.

Line 97, may be easily included -> can be, since Appendix D extends the proposed method, as I understand.

Line 284, clarify what figure in this phrase mean “treatment is each figure”.

Minor
Line 27 confouder, line 195 frm, line 336 DFIV

**Main Review:**

This work lifts the limitation of using linear or fixed kernel-based function classes in proximal causal learning, showing how to use neural networks which provide a more expressible and data-adaptive function class. The past approach based on two-stage regression is described in great detail, and the proposed modifications are clear. While the changing the method from static kernel-based features to adaptive neural network-based features is conceptually close to the approach of [32,18], the work demonstrates the significant technical effort required to make the change. The estimation method is shown to be consistent given assumptions on estimation of the regression functions in two stages and Rademacher complexity of the function classes used. The experiments show significant improvement from using neural networks.

My first concern is the artificial nature of the two experiments. The demand design task, while being realistic, simulates both the covariates and outcomes. Experiment on dSprite uses more realistic covariate distribution but the task itself (which is estimating the causal effect of an image on a synthetic outcome function) does not seem realistic or modelled after some real world application. I recognise that these are good benchmarks for testing the proposed methods that account for high-dimensional features. I would have appreciated demonstration of the method on a real world dataset or a synthetic one modelled after a task of practical interest and designed using realistic data distribution (in contrast to the demand design task).

My second concern is the limited motivation provided for the confounded off-policy evaluation problem which seems to be of practical value. But the description and possible applications of the problem is not very detailed. This can be improved by adding more example applications which are suitable to be modelled as confounded off-policy evaluation problem.

Questions:

Experiment setup:
How did the authors control the degree of complexity of confounder-treatment/outcome or outcome-treatment functions and dimensions of the three? Is it the case that Demand task has a  complex outcome function while dSprite task has complex confounder- treatment/outcome functions.

The method assumes continuous treatments, line 88, whereas in many practical settings we are concerned with quantifying treatment effect between two levels of the treatment. Does it extend to binary treatments? If not, then what is the difficulty on handling binary treatments?

Can you please provide example applications where the complexity of the datasets necessitate adaptive features?

Theoretical results:
The consistency results are left in terms of the Rademacher complexity of the function classes used. However, to justify the use of neural networks through these results, could you please comment on the Rademacher complexity of possibly a restricted class of neural networks? For example, a 2 layer ReLU network https://arxiv.org/abs/1805.12076.
Additionally, please discuss whether the Rademacher complexities can be shown to vanish and at what rate, even in some restricted class of neural networks.
Without these, the theoretical results are disconnected from the proposed method.

Method:
Can deep kernel learning (e.g. https://arxiv.org/abs/1511.02222) be a practical way of achieving adaptivity to data in the kernel-based methods [3,18]? Are there difficulties in learning such kernels for causal effect estimation tasks which in turn justify the proposed approach?

---
Author response addressed my concerns on experiments on realistic dataset and ideal application settings.

**Time Spent Reviewing:**

5

---

> ### Author Response · Authors · 2021-08-10
> **Response to Reviewer bAic**
>
> Thank you for the careful reading, and we will incorporate the suggested changes regarding wording and clarity.
> Below, we will address the major concerns and questions raised in the review.
> ---
>
> **Question 1:**
>
> > My first concern is the artificial nature of the two experiments. The demand design task, while being realistic, simulates both the covariates and outcomes. Experiment on dSprite uses more realistic covariate distribution but the task itself (which is estimating the causal effect of an image on a synthetic outcome function) does not seem realistic or modelled after some real world application. I recognise that these are good benchmarks for testing the proposed methods that account for high-dimensional features. I would have appreciated demonstration of the method on a real world dataset or a synthetic one modelled after a task of practical interest and designed using realistic data distribution (in contrast to the demand design task).
>
> **Answer:** Thank you for the suggestion. We will include the experiments on the real-world data studied in [3], in which the performance of DFPV matches the performance of KPV and PMMR. Please refer to the "official comment to all reviewers" for details.
>
> **Question 2:**
> > My second concern is the limited motivation provided for the confounded off-policy evaluation problem which seems to be of practical value. But the description and possible applications of the problem is not very detailed. This can be improved by adding more example applications which are suitable to be modelled as confounded off-policy evaluation problem.
>
> **Answer:** Thank you for the suggestion. We will include the following motivation in the introduction. In the flight ticket sales prediction example, a company might be interested in predicting the effect of discount offers. This requires us to solve the policy evaluation with $C=A$ since a new price is determined based on the current price. Or, if the company is planning to introduce a new price policy that depends on the fuel cost, we need to consider the policy evaluation with $C=Z$, since the fuel cost can be regarded as the treatment-inducing proxy variable.
>
> **Question 3:**
>
> > Experiment setup: How did the authors control the degree of complexity of confounder-treatment/outcome or outcome-treatment functions and dimensions of the three? Is it the case that Demand task has a complex outcome function while dSprite task has complex confounder- treatment/outcome functions.
>
> **Answer:** Yes, Demand task tests whether the method can learn a complex structural function, while dSprite task tests whether the methods can handle high-dimensional proxy variables.
>
> **Question 4:**
>
> > The method assumes continuous treatments, line 88, whereas in many practical settings we are concerned with quantifying treatment effect between two levels of the treatment. Does it extend to binary treatments? If not, then what is the difficulty on handling binary treatments?
>
> **Answer:** When treatments are binary (or discrete), we can use the one-hot vector representation (i.e. $\psi_A(0) = (1, 0)^\top, \psi_A(1) = (0, 1)^\top$) in DFPV. Although features for treatment are no longer adaptive, it is still beneficial to learn adaptive features for the proxy variables.
>
> **Question 5:**
> > Can you please provide example applications where the complexity of the datasets necessitate adaptive features?
>
> **Answer:** In the medical application, proxy variables can be high-dimensional. Say, we want to estimate the effect of a new chemotherapy treatment on a cancer. The CT image used in targeting treatment would be a good candidate for the treatment-inducing proxy.  DNA sequences might serve as the outcome-inducing proxy, given they are independent of CT scans or the application of medicine.  In this setting we need neural networks to model the bridge function because the proxy variables are structured and high-dimensional.
>
> **Question 6:**
> > Theoretical results: The consistency results are left in terms of the Rademacher complexity of the function classes used. However, to justify the use of neural networks through these results, could you please comment on the Rademacher complexity of possibly a restricted class of neural networks? For example, a 2 layer ReLU network https://arxiv.org/abs/1805.12076. Additionally, please discuss whether the Rademacher complexities can be shown to vanish and at what rate, even in some restricted class of neural networks. Without these, the theoretical results are disconnected from the proposed method.
>
> **Answer:** Thank you for the suggestion. We are happy to include the reference on vanishing Rademacher complexity of ReLU networks. However, it might be not straightforward to derive the Rademacher complexity of the specific class of neural networks we use, because we are employing outer product of two neural networks in Eq.(5,6), whose Rademacher complexity remains a challenging open problem.
>
> **Question 7:**
>
> > Can deep kernel learning (e.g. https://arxiv.org/abs/1511.02222) be a practical way of achieving adaptivity to data in the kernel-based methods [3,18]? Are there difficulties in learning such kernels for causal effect estimation tasks which in turn justify the proposed approach?
>
> **Answer:** Deep kernel learning uses a kernel function represented as $k(\mathbf{g}(x; w), \mathbf{g}(y;w))$, where $k$ is a base kernel function and $\mathbf{g}(x; w) \in \mathbb{R}^d$ is a neural network parameterized by $w$. In this sense, DFPV can be interpreted as deep kernel learning, where $\mathbf{g}$ corresponds to feature maps $\mathbf{\phi}_A, \mathbf{\psi}_A$ and base kernel is the ordinary inner product $k({x}, {y}) = {x}^\top {y}$. We may use more complex base kernels (e.g. RBF kernel), but this requires a significant amount of computation, since it involves inverting a Gram matrix, with complexity cubic in the data size. By contrast, DFPV runs in linear time with the data size, since we only invert matrices size of $d \times d$ in Equation (9).

---

> > ### Comment · Reviewer_bAic · 2021-08-28
> > **Thanks for the clarifications**
> >
> > I thank the authors for providing experimental results on a realistic dataset (Grade retention) and clarifying the application areas where adaptive features may help. My concerns are adequately addressed. Hence, I have increased my score to 7 (Good paper, accept).
> >
> > I concur with other reviewers that there should be more discussion on limitations such as need for more data for DFPV and as yet unverifiable technical conditions. I also encourage authors to discuss limitations of the consistency analysis in its use of Rademacher complexity for the proposed class of neural networks.

---

### Author Response · Authors · 2021-08-10
**Comments to all reviewers (rebuttal)**

All reviewers suggested to have additional experiments. We will therefore add the following two experiments, which address the purposes raised by the reviewers:
1. Synthetic setting studied in [18]

    To test the performance of DFPV in a simpler setting, we will add the result based on the synthetic setting studied in [18]. This has a single-dimension treatment variable and two-dimensional treatment-/outcome- proxy variables. The outcome variable is generated from a cosine function, which is smooth enough to be modeled by kernel functions. Under this setting, DFPV performs slightly worse and has a larger variance than KPV and PMMR. (See the plots for the result at <https://www.sendspace.com/file/etfsbd>) This is not surprising, since DFPV tends to require more data than KPV and PMMR, as needed to learn the neural net feature maps (rather than using fixed pre-defined kernel features). Hence, we can say that we should favor KPV and PMMR over DFPV when the data is low-dimensional and the relations between the variables are smooth. We would like to note, however, DFPV outperforms CEVAE, which shows that the proxy setting is still required.

2. Grade Retention dataset [3]

    To test the performance of DFPV in a more realistic setting, we will add an experiment on the Grade Retention dataset introduced by [3].  This aims to estimate the effect of grade retention based on the score of math and reading on the long-term cognitive outcomes, in which we use scores in elementary school as a treatment-inducing proxy (Z)  and  cognitive test scores from Kindergarten as the an outcome-inducing proxy (W). Following [18], we generate a synthetic "ground truth" by fitting a generalized additive model to learn a structured causal model (SCM), and a Gaussian mixture model to learn unmeasured confounder based on the learned SCM (this is needed since for real-world data there is no measured ground truth). In this setting, the performance of DFPV matches KPV and PMMR.(MAE of DFPV: $0.026 \pm 0.006$ for math and $0.025 \pm 0.006$ for reading.) As in the previous experiment, the setting is low-dimensional (one-dim treatment variable, three-dim treatment-inducing proxy, four-dim outcome-inducing proxy) and the generative model is smooth (the "ground truth" being a generalized additive model and a Gaussian mixture model). For these reasons, we might again expect this data to favor kernel methods, such as KPV and PMMR; nonetheless, our method matches them.  DFPV again outperforms CEVAE (MAE of CEVAE: $0.051 \pm 0.023$ for math and $0.052 \pm 0.023$) in this setting.

---

### Decision · Program_Chairs · 2021-09-27

**Decision:**

Accept (Poster)

**Comment:**

The expert reviewers all appreciated the paper and agree it provides a useful new method and that the paper should be accepted. The reviewers appreciated the authors' response which better highlighted the significance of the contribution, including the relevant applications and what is lacking in previous work. The authors are expected to address the points raised by reviewers in a final version, including incorporating the additional clarifying discussion in their responses.